# Modelling chemical processes in explicit solvents with machine learning potentials

Hanwen Zhang ®[1], Veronika Juraskova ®[1] & Fernanda Duarte ®[1] ✉

Solvent effects influence all stages of the chemical processes, modulating the stability of intermediates and transition states, as well as altering reaction rates and product ratios. However, accurately modelling these effects remains challenging. Here, we present a general strategy for generating reactive machine learning potentials to model chemical processes in solution. Our approach combines active learning with descriptor-based selectors and automation, enabling the construction of data-efficient training sets that span the relevant chemical and conformational space. We apply this strategy to investigate a Diels-Alder reaction in water and methanol. The generated machine learning potentials enable us to obtain reaction rates that are in agreement with experimental data and analyse the influence of these solvents on the reaction mechanism. Our strategy offers an efficient approach to the routine modelling of chemical reactions in solution, opening up avenues for studying complex chemical processes in an efficient manner.

(Bio)chemical and industrially relevant reactions occur predominantly in the liquid phase[1,2]. The influence of solvent on reaction rates was first documented by Berthelot and Pean de Saint-Gilles in 1862[3] and later formalised by Menschutkin in 1890[4]. Since then, experimental and computational techniques have been developed to elucidate and quantify the origin of solvent effects on reactivity and selectivity[5]. From an atomistic point of view, solvent effects arise from the interactions between solute and solvent molecules, which, although generally weak, have a significant impact on the overall reaction dynamics.

In computational chemistry, solvent effects are modelled using implicit or explicit solvent models. The former represents solvents as a polarisable continuum, offering computational simplicity and efficiency. However, it fails to capture the contributions arising from solute-solvent interactions, including entropy and pre-organisation effects[6]. On the other hand, explicit solvent models provide an atomistic representation of the solvent but at a much higher computational cost since they require the use of ab initio molecular dynamics (AIMD). This cost becomes significant when attempting to compute free energies, where extensive sampling is required to obtain statistically meaningful ensembles. Hybrid approaches, such as quantum mechanics/molecular mechanics (QM/MM), can alleviate this computational cost by describing only the reactive part at a QM level while the environment is described classically. However, the inclusion of mobile solvent molecules into the QM part creates additional technical difficulties, such as discontinuities at the boundary between the QM and MM regions[7,8].

In recent years, machine learning-based potentials (MLPs) have emerged as powerful surrogates for existing modelling techniques, representing complex potential energy surfaces (PES) with an accuracy comparable to QM methods but at a significantly lower computational cost[9,10]. Since the pioneering work of Behler and Parinello (BP) on neural network-based potentials (NNP)[9], several MLP approaches have been developed, including models directly derived from BP, such as DeepMD[11] and ANI[12], message-passing NN models[13], such as PhysNet[14], SchNet[15], and MACE[16], kernel-based approaches, such as Gaussian Approximation Potential (GAP) and gradient-domain ML (GDML)[17–19], and linear regression-based models, such as linear Atomic Cluster Expansion (ACE)[20,21]. These MLPs have been successfully applied to study organic molecules[12,22], material properties[23,24] and simple chemical reactions in the gas phase and interfaces[25–28]. While there are prominent examples of using MLPs to correct QM/MM computed free energies[29], modelling solvent implicitly[30,31], or replacing the QM part to have a more efficient approach, ML/MM[32], only a handful of examples exist where chemical processes have been modelled in explicit solvent fully using MLPs. Such examples include urea decomposition in water[33], 1,3-dipolar cycloadditions in water[34], Strecker-cyanohydrin

[1]Chemistry Research Laboratory, Oxford, United Kingdom. ✉e-mail: fernanda.duartegonzalez@chem.ox.ac.uk

synthesis of glycine in water[35], and reactions in alkali carbonate-hydroxide electrolytes[36]. In all these cases, NNPs are used, requiring thousands of AIMD configurations. For a comprehensive overview of relevant works, we refer the reader to a recent review by Isayev and Roitberg, ref. 37, and SI §S1.

One of the main challenges limiting the application of MLPs to study chemical reactions in solution is the quality and size of the training data set, which needs to incorporate diverse configurations to effectively capture solute-solvent interaction at the minima and TS regions. Unlike solid-state systems, where particles are packed in highly ordered arrangements, molecules in solution are often flexible, resulting in a larger conformational space that needs to be sampled to accurately describe their PES. Moreover, the reliable description of reaction barriers and non-covalent interactions requires accurate reference electronic structure methods, further increasing the computational cost[38]. The complexity of the training data can be reduced in Δ-ML models[39], which predict the differences between semi-empirical baseline and QM PES rather than the absolute energies. However, even in these approaches, the cost of energy and force evaluations remains a bottleneck, as the baseline computations need to be performed in each MD step[40,41].

Recently, we demonstrated the promise of several MLP approaches, including ACE, GAP and NequIP, in modelling various molecular systems and chemical reactions[42,43]. Notably, when coupled with active learning (AL), where the MLP is retrained using new data collected based on preliminary versions of MLP, these methods can provide accurate potentials with much smaller data sets than traditional NN approaches. In this way, one can efficiently explore the chemical space required to describe reactions involving complex PESs.

The performance and efficiency of AL strategies heavily depend on the algorithm used to select suitable structures for retraining. Previous studies have utilised various metrics or selectors to evaluate the performance of MLPs within AL iterations. Probably the most straightforward metric is the direct comparison of energies or forces between the reference (ground-truth) method and the MLP[42]. However, this metric requires QM calculations at every evaluation step, which is only feasible for small systems in the gas phase. Gaussian Process-based methods, such as GAP, commonly use energy uncertainty as a metric, which refers to the variance in predicted energy[28,44–46]. However, this property is limited to Bayesian models, as they yield a predictive distribution characterised by its mean and variance. Other models only provide a point prediction for a given structure and are thus unable to utilise this metric[47]. Another approach widely used in NNPs[34,48–50], although not limited to, is query-by-committee, where uncertainty is quantified through the variance of predicted energies or forces among an ensemble of MLPs, referred to as committees. In the case of NNs, multiple potentials are trained with the same architecture but with different initial parameters. This approach provides on-the-fly uncertainty estimation but incurs additional computational costs associated with the training of several NNPs. In all the uncertainty metrics mentioned above, a large variance indicates an under-representation of the corresponding data points and the need to include them in the training set.

In this work, we propose a general AL strategy for training MLPs to model chemical processes in explicit solvents and, more generally, large systems with complex PES. To achieve this, we employ an AL loop combined with descriptor-based selectors. In contrast to the metrics described above, these selectors use molecular descriptors, such as Smooth Overlap of Atomic Positions (SOAP)[51], to assess whether the training set accurately represents the chemical space of interest. By examining the SOAP descriptor space for the structures in the training set, we demonstrate that these selectors provide a general metric applicable across different MLP approaches at a low computational cost. The effectiveness and versatility of the selectors are showcased using the linear ACE model. Firstly, we investigate a simple water box

as a test case, evaluating and comparing the performance of various selectors in terms of the quality of the potential and data efficiency. Building upon these findings, we further apply our strategy to study the Diels-Alder (DA) reaction between cyclopentadiene (CP) and methyl vinyl ketone (MVK) in both water and methanol. Through this example, we illustrate the transferability of our approach to accurately model chemical reactions in different solvent environments. The generality and computational efficiency of our approach make it well-suited for a wide range of applications, contributing to the advancement of MLP in various fields. By harnessing its capabilities, accurate and efficient modelling of chemical processes in explicit solvents becomes attainable. This enables a deeper understanding of solvent effects and facilitates the investigation of more complex processes in solution.

## Results
### Workflow
The AL strategy employed in this work to train MLPs is illustrated in Fig. 1a. The first step involves generating a small set of configurations labelled with reference energies and forces. This information is used to train the initial version of the MLP. For a given reaction, we employ two different training sets, one containing the reacting substrates in the gas phase (or implicit solvent) and another including explicit solvent molecules. The latter is necessary to account for specific non-covalent interactions between the solute and solvent.

Different initial data generation strategies are used for each of the data sets mentioned above. For the gas phase or implicit solvated molecules, initial training configurations are generated by randomly displacing the atomic coordinates. In the case of a chemical reaction, the training typically starts from the corresponding TS. The initial dataset containing substrate and the explicit solvent can be generated either from solvent molecules in a box under periodic boundary conditions (PBC) or cluster models containing only a handful of solvent molecules placed at relevant positions. While PBC reproduces well the structure of the bulk solvent and includes long-range interactions, generating such training data, in particular with AIMD, is computationally expensive as thousands of configurations are required to accurately describe all the interactions, making it unfeasible for most chemically relevant systems. An alternative approach for generating PBC data is to use classical MM force fields; however, they are often inaccurate and unable to describe bond-breaking/forming processes. Moreover, MM configurations exhibit a weak overlap with the true potential energy surfaces (PES), even for non-reactive systems making them unsuitable for the training of MLPs[42]. In this regard, cluster data labelled with molecular energy and forces provide all structural information for MLPs based on the local descriptors and offer access to a large spectrum of electronic structure methods, including long-range corrected and double-hybrid DFT functionals. The minimum radius of the solvent shell around the substrate should be no less than the cut-off radius used for training the MLP to avoid artificial forces close to the solvent-vacuum interface in the cluster data.

As discussed in more detail below, we observed good transferability of cluster-based MLPs to systems with PBC. Similar transferability has already been reported for an NNP applied to bulk water, where NNPs trained using both PBC and cluster training data demonstrate similar performance in predicting bulk properties, such as radial distribution functions (RDF), self-diffusion coefficients, and equilibrium densities[52].

After the initial MLP training, one structure from the initial training set is selected as the starting point to propagate the molecular dynamics (MD) using the first version of the trained MLP. Several rounds of short MD simulations are then performed to assess the stability of the potential and generate new training structures. The simulation time is set to $(n^3 + 2)$ fs where $n$ corresponds to the index of the MD run, starting from 0. The time step of the dynamics is 0.5 fs.

**a** Active learning workflow

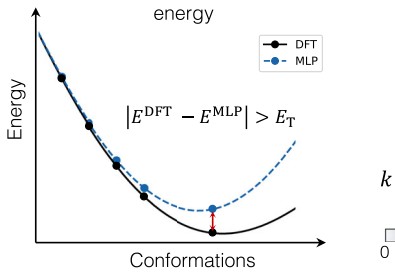

**b** Selectors

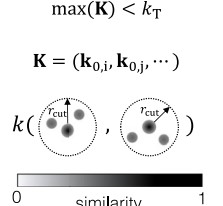

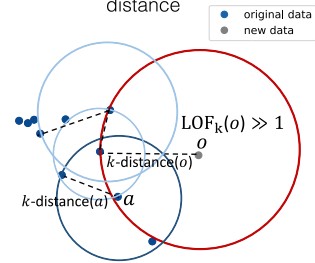

**Fig. 1 | Active learning (AL) workflow and structure selection strategies for training a machine learning potential (MLP). a** In the gas phase, initial configurations are generated by random displacements ($\Delta \mathbf{D}$) on the atomic coordinates ($x, y, z$, red arrows). For the system in the condensed phase, cluster configurations with a minimum radius ($r_{cluster}$) ≥ cut-off hyperparameters ($r_{cut}$) in MLP are used as initial configurations. **b** Selection methods in the AL process: *energy* selector: selects data points where the energy difference (red) between the ground truth density functional theory (DFT) value ($E^{DFT}$) and MLP predicted values ($E^{MLP}$) exceeds the threshold ($E_T$). *similarity* selector: collects configurations with dissimilar geometry to the original data set, i.e., configurations with the maximum of the similarity vector **K** lower than similarity threshold $k_T$. *distance* selector: uses the local outlier factor (LOF) method to identify outliers $o$ from the original data set. The circle with radius $k$-distance centred at the outlier ($o$) is highlighted in red.

More details can be found in ref. 42 and SI§ S6.1. From each MD trajectory, the last frame is evaluated by the selector to determine whether to add or not this structure to the training set. If the structure is not selected, $n$ is incremented, and MD runs are repeated until the maximum simulation time or a maximum number of MD iterations is achieved.

### Descriptor-based selectors

In previous studies, we have utilised an energy-based selector (further referred to as *energy*) to determine whether a configuration should be added to the training set. This selector identifies structures that show an error in predicted energy higher than the threshold $E_T$, satisfying the condition $|E^{DFT} - E^{MLP}| > E_T$. Structures with prediction errors greater than $10\,E_T$ were excluded from the dataset as these were too distorted to provide meaningful information. Although reliable, this approach requires QM calculations for each selection step, which is computationally prohibitive for large systems.

In this work, we introduce selectors using descriptor-based selection criteria. Descriptors transfer molecular representation from Cartesian coordinates to a physical invariant description, encompassing both the geometrical and chemical information of molecular structures. Evaluating the SOAP descriptor over the training data thus provides information on how well the training set covers the relevant conformational and chemical space, enabling the identification of underrepresented data points.

During the AL loop, we apply so-called similarity- and distance-based selectors, referred here to as *similarity* and *distance* selectors, respectively. The former quantifies the similarity between a new data point $p$ and existing configurations $p'$ using the kernel function $k(p \cdot p')$. The similarity vector of the data point is defined as:

$$\mathbf{K} = \left( \left| k\left(\mathbf{p}_0 \cdot \mathbf{p}_i\right) \right|^{\zeta}, \left| k\left(\mathbf{p}_0 \cdot \mathbf{p}_j\right) \right|^{\zeta}, \cdots \right)^T \quad (1)$$

where $\mathbf{p}_0$ is the SOAP vector of the new structure, $\mathbf{p}_i$ is the SOAP vector of the $i$-th configuration in the existing set, and $\zeta$ is a positive integer that increases the sensitivity of kernel to changes in atomic position[51]. The kernel is computed between the new configuration and all other configurations in the training data set. The selector adds structure to the training set if the maximum value of its similarity vector, **K**, is smaller than threshold $k_T$, i.e., max(**K**)$<k_T$. Selecting an appropriate threshold is key as too low values (e.g., similarity below 0.9) can result in the selection of non-physical structures that fail to converge in the self-consistent field (SCF) computations, while too high values (e.g., 1) do not provide any additional information.

For the *distance* selector, we use the local outlier factor (LOF) method[53] to determine whether the SOAP vector of the new configuration is an outlier compared to the SOAP vectors of the existing training data. LOF is based on the local density of each point, which is calculated by measuring the Euclidean distance between the target point and its $k$-nearest neighbours (Fig. 1b). The local reachability density of an object $o$, denoted as $lrd_k(o)$, is calculated as:

$$lrd_k(o) = \left( \frac{\sum_{i \in N_k(o)} rd_k(o, i)}{|N_k(o)|} \right)^{-1}. \quad (2)$$

$rd_k(o, a)$ in the equation corresponds to the reachability distance defined as

$$rd_k(o,a) = \max(k\text{-distance}(a), d(o,a)). \quad (3)$$

Here, $k$-distance $(a)$ represents the radius of the smallest circle with its origin in $a$, which includes the $k$-nearest neighbours of $a$ (illustrated in dark blue in Fig. 1b), $d(o, a)$ is the Euclidean distance between points $o$ and $a$, where point $o$ is the target point and point $a$ is one of its neighbours. $N_k(o)$ is a set of $k$-nearest neighbours of $o$, which are illustrated by blue points. The local reachability density is thus expressed as the number of neighbours per distance unit. If the local reachability density of the target point is smaller than that of its neighbours, the point is considered an outlier and added to the training data set. The comparison among the local densities is achieved by computing the ratio of the average local density of neighbours and the local density of the point, as follows:

$$LOF_k(o) = \frac{\sum_{i \in N_k(o)} \frac{lrd_k(i)}{lrd_k(o)}}{|N_k(o)|} \quad (4)$$

A LOF value close to 1 indicates that a point is located in a similarly dense region as its neighbours. A LOF value less than 1 represents an inlier, meaning that the point is situated in a denser region, while a value greater than 1 indicates an outlier. Since there is no definitive rule for selecting an LOF threshold to identify outliers, in this study, we chose the threshold as an LOF value which is larger than 80 % of LOF values for the given training set. This approach ensures that the threshold value varies with the PES exploration during the AL iterations. More details of the *distance* selector are provided in SI§ 3.2.

In contrast to the variance metric, descriptor-based selectors can be applied to any regression method without incurring additional computational costs for training multiple models.

As discussed in the next section, both descriptor-based selectors accelerate the training of MLP models compared to the *energy* selector. This acceleration arises from the reduction in QM calculations and the selectors' ability to explore the relevant chemical space more efficiently.

### Performance of selectors - water models
We assessed the performance of three different selectors (*energy*, *similarity*, and *distance*) during the AL training of MLPs for water. In each case, we evaluated the accuracy of the generated potential by measuring the mean absolute deviation (MAD) of total energy and atomic forces with respect to the ground truth method, PBE0-D3BJ. We also considered the efficiency of training by analysing the number of configurations required for training. The AL process was considered complete when no configurations were selected within the maximum simulation time of 5 ps.

Our results demonstrate that, in general, all descriptors provide accurate and stable potentials. This is evident from the direct comparison of predicted and ground-true energies for two systems: a small cluster system comprising 27 water molecules, from which we extracted 200 configurations from a 1 ps MD simulation, and a larger system consisting of 216 water molecules under PBC, where we performed 50 ps dynamics in the NVE ensemble (SI § S3.1). All MLPs achieved MAD errors in energy below 1 kcal mol$^{-1}$ and errors in force of less than 2 kcal mol$^{-1}$ Å$^{-1}$. Moreover, all MLPs remained stable during the NVE simulations, which lasted significantly longer than the maximum simulation time in the AL process.

The main difference between the selectors lies in the number of configurations selected during the training and computational efficiency. Descriptor-based selectors generally require fewer configurations, with 40 and 52 data points for *similarity* and *distance* selectors, respectively, compared to 281 data points for *energy*. To explore the

geometrical similarity among the data selected by different selectors, we combined all the training sets (373 configurations in total, shown in grey in Fig. 2a) and analysed them with t-SNE maps based on global SOAP representation. This analysis reveals three distinct clusters. The bottom cluster represents initial structures in training, where water molecules are randomly placed and have relatively high energy. The middle cluster contains configurations with more structured arrangements due to the presence of hydrogen bonds (HB). Finally, the upper cluster comprises configurations near equilibrium, evidenced by lower energies. Interestingly, the energy selector predominantly selects geometries near the equilibrium configurations (67%), with only a handful of configurations in the middle/bottom cluster. In contrast, the configurations selected by either the *similarity* or *distance* selectors are more evenly distributed (Fig. 2b). The size of the training set does not influence the MLP performance, as all MLPs reproduced well the experimental radial distribution function (RDF) of water (Fig. 2c). The small differences in the positions and amplitudes of peaks for the first and second solvation shells are likely due to the level of theory used and the lack of nuclear quantum effects[54].

This demonstrates that both *similarity* and *distance* selectors explore the chemical space more efficiently, using only around 15% of the training data required by the *energy* selector. Furthermore, they reduce the structural correlation in the training sets, as illustrated in Fig. 2a. Instead of containing numerous points with similar geometries, these training sets encompass data points distributed across the space. Descriptor-based selectors are also significantly faster than energy-based ones as they do not require QM computation at every selection step, making them suitable for larger and more complex systems.

Overall, the descriptor-based selectors demonstrate superior efficiency in terms of training speed and amount of data, outperforming the *energy* selector. All three selectors yield potentials with comparable accuracy and stability. However, the *distance* selector requires a more extensive initial data set to perform the selection criteria in the first iteration of the AL process, as it relies on neighbouring information. Throughout this paper, we will use the *similarity* selector for MLP training.

### Training strategy - DA reaction of CP and MVK in explicit water
The ability of MLPs to accurately describe chemical processes in explicit solvents is critical for extending their application to more challenging systems. To this end, we investigated the solvent effects on the DA reaction between CP and MVK in explicitly modelled water and methanol. While this reaction exhibits only minor solvent effects compared to charged systems, several reports have shown rate acceleration and selectivity enhancement when water or aqueous solvent mixtures are employed[55–58]. The reaction is accelerated up to 58-fold in water compared to methanol[56], and the *endo/exo* selectivity is enhanced by 8-fold in water over benzene[58]. This behaviour is widely explained by the formation of stronger HBs between solvent and substrate at the TS compared to the reactants state (RS) and product state (PS)[59–65]. In addition, solvent polarity and hydrophobic effects have also been suggested to contribute to this enhancement[55,62,66,67].

We trained four ACE MLPs for each *endo/exo* DA reaction between CP and MVK in either explicit water or methanol. For comparison, we also trained the models in implicit solvent, neglecting the presence of the explicit solvent molecules. Here, we describe the strategy employed to obtain the ACE MLP for the *endo* reaction in explicit water. A similar approach was used for the other systems (SI § S6 for further details). For the reaction in explicit solvents, the training set consisted of four subsets, each aimed at describing different types of interactions in the system. Subset 1 corresponds to the substrate complex (CP + MVK) and provides information about intramolecular interactions and intrinsic reactivity (Fig. 3a). Subsets 2 and 3 consist of the substrate with 2 and 33 water molecules, respectively, and aim at describing various solute-solvent interactions. Finally, subset 4

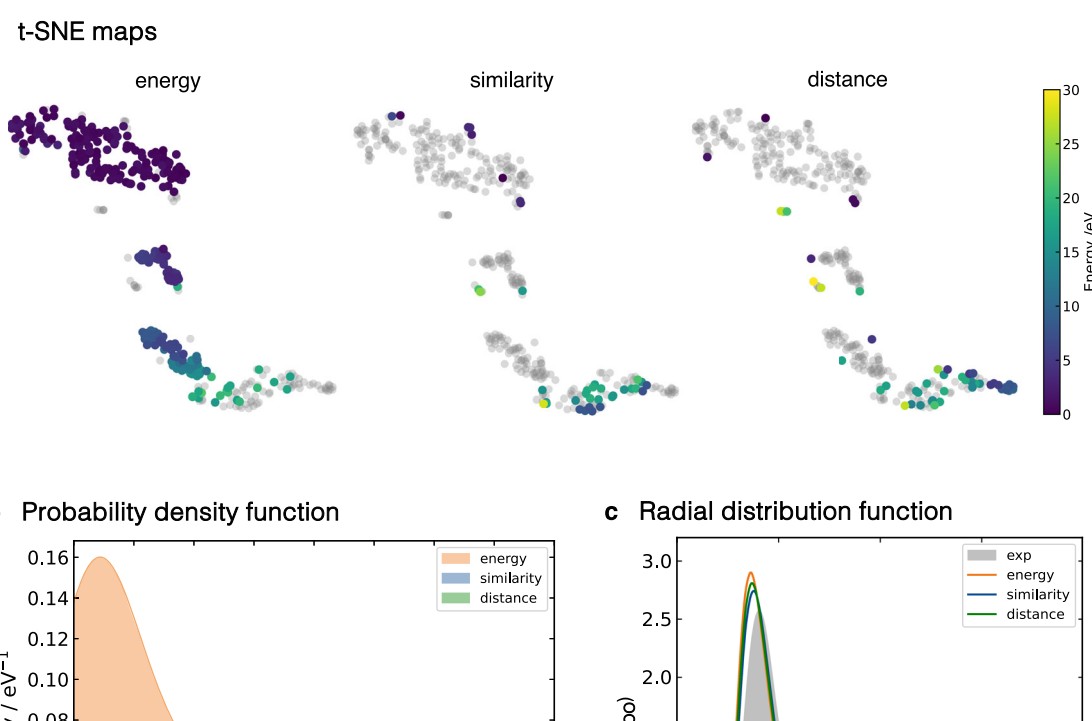

**Fig. 2 | Comparative analysis of selectors. a** t-Distributed Stochastic Neighbour Embedding (t-SNE) maps of configurations generated during active learning using *energy*, *similarity* and *distance* selectors for a 27-water molecule system. Configurations generated by each selector are labelled by their energy relative to the lowest energy configuration in the dataset, while configurations obtained from other selectors are shown in grey. **b** Energy distribution of configurations obtained using the different selectors for a 27-water cluster system. **c** Oxygen-oxygen radial distribution function (RDF) from a 20 ps NVT Atomic Cluster Expansion (ACE) machine learning potential molecular dynamics (MLP-MD) simulation of 216 water molecules in an 18.65 Å cubic box under periodic boundary conditions. RDF obtained from ACE MLP trained with *energy* (orange), *similarity* (blue), or *distance* (green) selector. Experimental RDF is shown in grey shading[88]. Data associated with this figure is provided as a Source Data file.

contains only water molecules, providing information about solvent-solvent interactions in bulk solvent. All subsets were generated independently using the AL scheme initiated from the transition state structure, except the pure water subset, which started from a random water configuration. This approach ensured that the training set contained reactants, products and connecting reaction paths for all studied environments. The combination of these sub-training sets yielded 600 training points, which we used to train the final ACE MLP. See SI § S6.1 for further details.

The accuracy of the resulting ACE MLP was assessed by conducting 500 fs ACE MLP-MD simulations starting from two configurations not included in the training set. The first configuration, which is similar to subset 2, consisted of the gas phase TS and three water molecules forming HBs with the carbonyl group of MVK. The resulting dynamics are stable with the energy error of 2 meV atom$^{-1}$, demonstrating the ability of the ACE MLP to represent reactions and specific HB interactions (Supplementary Fig. 14). The second one corresponds to a TS immersed in a box containing 55 water molecules (box size: 12.42 Å, Fig. 3b). The resulting accuracy confirms that ACE MLP is reliable for investigating the reaction of CP and MVK in solution. Validation of the other systems is provided in the SI § S6.

### Application of ACE MLPs - DA reaction of CP and MVK in solvents

After obtaining accurate and stable ACE MLPs for the reaction of CP and MVK in two solvent environments, we utilised these potentials to investigate the reaction pathway in more detail. This was done by conducting a relaxed 2D scan along the forming of C-C bonds $r_1$ and $r_2$ in both implicit and explicit solvents (Fig. 4a and Supplementary Fig. 26, respectively). Analysis of the *endo* 2D scan in both implicit and explicit water reveals the presence of a zwitterionic-like structure characterised by the formation of only one C-C bond in the region around $r_1 < 1.6$ Å and $2.5 < r_2 < 3.0$ Å (an example marked by a cross in Fig. 4a). Unrestricted DFT calculations on these geometries confirmed that they do not exhibit any diradical character (SI § S8). The cross-labelled zwitterionic species is slightly more stabilised in the explicit solvent than in implicit solvent ($\Delta\Delta E = 2.2$ kcal mol$^{-1}$).

Using ACE MLP-MD in conjunction with umbrella sampling (US, ACE MLP-MD/US), we then computed the activation free energy, $\Delta G^{\ddagger}$, in implicit and explicit solvents (Fig. 4b and Supplementary Fig. 29). $\Delta G^{\ddagger}$ in implicit solvent are 21.2 kcal mol$^{-1}$ and 23.6 kcal mol$^{-1}$, for the *endo* and *exo* pathway, respectively ($\Delta\Delta G^{\ddagger} = 2.4$ kcal mol$^{-1}$). Incorporating explicit solvent reduces these values to 18.8 kcal mol$^{-1}$ and 20.3 kcal mol$^{-1}$ ($\Delta\Delta G^{\ddagger} = 1.5$ kcal mol$^{-1}$), thereby improving the agreement with experimental data (19.2 kcal mol$^{-1}$ and 21.1 kcal mol$^{-1}$,

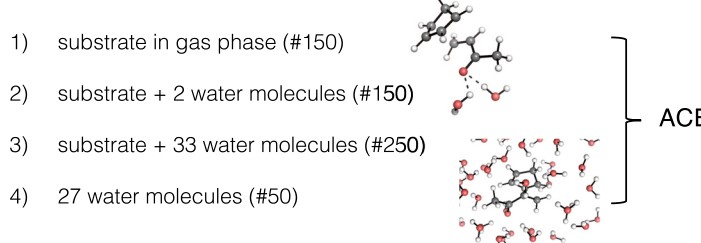

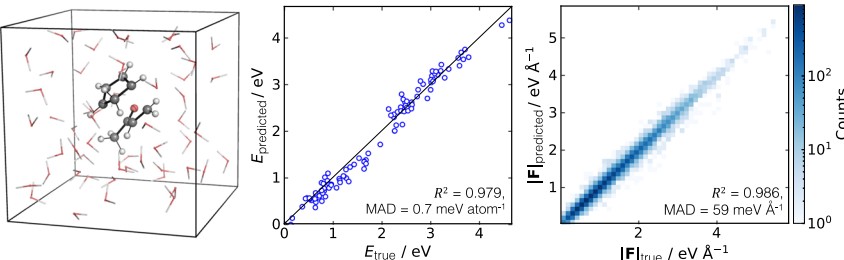

**a** Sub-training sets

1) substrate in gas phase (#150)

2) substrate + 2 water molecules (#150)

3) substrate + 33 water molecules (#250)

4) 27 water molecules (#50)

ACE

**b** Validation of ACE

**Fig. 3 | Training approach and accuracy of Atomic Cluster Expansion (ACE) machine learning potential (MLP) for *endo* Diels-Alder reaction of cyclopentadiene (CP) and methyl vinyl ketone (MVK) in explicit water. a** The training data consists of four subsets, each describing key interactions. **b** Comparison of ground-truth and ACE MLP energies and forces over a 500-fs ACE MLP-molecular dynamics (MD) downhill dynamics for a system containing substrate and 55 water molecules with a timestep of 0.5 fs at 300 K.

respectively[56,57,65], Supplementary Table 5). These results also illustrate the significance of solute-solvent interactions in the reaction.

Furthermore, the presence of solvent molecules influences the synchronicity of the reaction. In explicit solvent, the reaction exhibits an earlier and more asynchronous TS compared to the implicit solvent or gas phase. The difference in bond length between the forming C-C bonds $\Delta r = |r_2 - r_1|$ at the TS increases from 30 pm in the gas phase to 37 pm in implicit solvent and 46 pm in explicit water. Notably, explicit water molecules also altered the reaction mechanism from a concerted asynchronous to a "pseudo" stepwise mechanism, as evidenced by the presence of a shallow local minimum in the FESs immediately after the TS (Fig. 4b). This intermediate state, which was observed for both *endo* and *exo* reactions, corresponds to a zwitterionic state. Interestingly, such an intermediate is absent in the PES, where the structure corresponds to a high energy state (labelled as a cross in (Fig. 4a)). This behaviour suggests that the intermediate arises from an entropic rather than an enthalpic contribution. The formation of an entropic intermediate has been previously reported by Singleton et al. for the reaction of cis-2-butene with dichloroketene, in which the free energy surface illustrates the entropic barrier and the mechanism change from concerted to stepwise[68]. In this study, we observed a similar phenomenon, highlighting the necessity of explicit solvent to capture the formation of an entropic intermediate.

The presence of this entropic intermediate is further confirmed by downhill dynamics initiated from the TS (Fig. 4c and Supplementary Table 13). For the *endo* reaction, the trajectories reveal a significantly more asynchronous reaction in explicit solvent compared to implicit solvent, in agreement with the ACE-MD/US data. The asynchronicity in the downhill dynamic is evident in the increased average time gap between the formation of the two C-C bonds, from 19.9 fs in an implicit water solvent to 84.3 fs in an explicit solvent. It is worth noting that

although the average time gaps observed in the water exceeded 60 fs (the time gap criterion proposed by Houk et al. to distinguish concerted and stepwise mechanisms[69]), some trajectories displayed time gaps below this threshold, indicating that not all trajectories passed through the intermediate region, and certain trajectories bypass the intermediate free energy well. The presence of this intermediate did not affect the product ratio, as the intermediate lifetime was shorter than the C-C bond rotation period, leaving no time to form alternative products by bond rotation.

A comparison of the average time gap for the reactions in explicit water and methanol reveals that the latter exhibits a more concerted mechanism, with a time gap of 24.8 fs. These distinct reaction mechanisms are in line with the differences in the synchronicity of this reaction in different solvents, where the TSs exhibit a $\Delta r$ of 7 pm in methanol and 46 pm in water (listed in Supplementary Table 12). In addition, a shift in the stability of the intermediate species occurs in methanol (further discussed in SI § S10.1).

To assess the effect of solvent at the molecular level, we performed uphill trajectories propagated from the optimised RS towards the TS to PS (further details in SI § S10.2). In contrast to downhill dynamics propagated from the TS, uphill dynamics allow the solvent sufficient time to reorganise before the trajectory passes the free energy barrier, providing a more realistic view of solvent behaviour during the reaction.

To determine the importance of HB stabilisation throughout the reaction, we analysed the number of HBs as well as their bond lengths (O(carbonyl)-H(water)) and angles (O(carbonyl)-H(water)-O(water)) distributions at RS, TS and PS in explicit water and methanol (Supplementary Table 15 and Supplementary Fig. 34). We also analysed the density distribution of water molecules surrounding the reactive species at the RS and intermediate states obtained from the

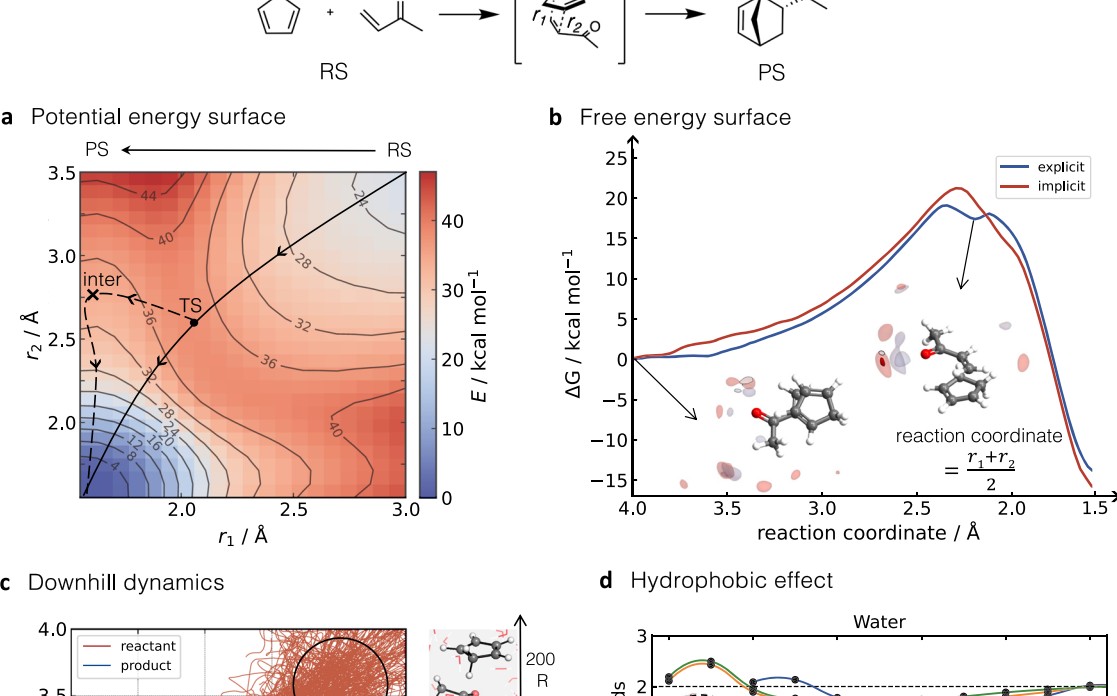

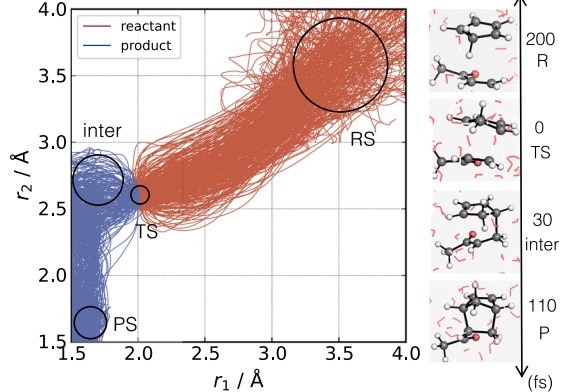

**Fig. 4 | Diels-Alder reaction of cyclopentadiene (CP) and methyl vinyl ketone (MVK) with $r_1$ and $r_2$ representing distances of the two formed C-C bonds. The reactant state (RS), transition state, and product state (PS) are depicted. a** 2D potential energy surface (PES), which is along $r_1$ and $r_2$ distances, generated by Atomic Cluster Expansion (ACE) machine learning potential (MLP) in explicit water (box size 18.5 Å). Solid and dashed lines indicate the reaction pathway from PES and ACE MLP-molecular dynamics (MD), respectively. **b** Free energy surfaces (FES) obtained from ACE MLP-MD/umbrella sampling (US) simulation along the reaction coordinate of $(r_1 + r_2)/2$. The distribution of oxygen (red) and hydrogen (grey) atoms around the solute is shown for the reactants and intermediate state. **c** Downhill ACE MLP-MD dynamics in explicit water along $r_1$ and $r_2$ for 500 trajectories, including snapshots of forward/backward trajectories. **d** Number of hydrogen bonds per solvent molecule during explicit solvent uphill trajectories at the RS, intermediate (inter) if existent, and PS states as a function of distance ($D$) from the centre of mass (CoM) of the reactive molecules. Data associated with this figure is provided as a Source Data file.

corresponding US windows (Fig. 4b). The latter was chosen over the TS due to their similar geometry and the fact that only a few configurations representative of the TS were obtained. In both reactions, the number of HBs and the density of water's oxygen and hydrogen atoms around the substrate remained practically constant. However, the reaction in water exhibited stronger HB interactions compared to methanol during the reaction process, as evidenced by the existence of HBs with shorter bond lengths at TS in water and a larger decrease in the angle in methanol.

These observations challenge the hypothesis that the reaction is accelerated in water due to the stabilisation of the TS by enhanced HB interactions compared to the RS, as suggested in previous studies[59,60,70]. These differences can be attributed to the differing dynamics employed in the studies. For example, Houk et al. investigated the reaction in water using downhill dynamics from the TS and observed shorter HBs bond length and more linear bond angle at TS compared with RS and PS. In this approach, the TS with solvent

molecules is fully optimised, and the solvent does not undergo complete reorganisation at RS or PS as the trajectory approaches the RS or PS at a faster rate than the reorganisation process due to the absence of energy barriers to overcome.

The reactions in solution are further affected by hydrophobic effects[66], which were investigated by analysing the change in cavity volume during uphill dynamics. The cavity volume was measured as empty space formed by the solvent when the solute was extracted from the system. Supplementary Fig. 33 illustrates the reduction in cavity volume from the RS to the PS. Reduction in the cavity is more pronounced in water with a change in cavity volume from RS to post-TS of −60 Å³ compared to methanol (−40 Å³). The decrease in cavity volume is entropically unfavourable as it implies the formation of a more ordered solvent structure. However, at the same time, it is enthalpically favourable due to the formation of HBs among solvent-solvent (rather than solvent-solute) molecules, resulting in a decrease in $\Delta G^{\ddagger}$. The importance of HB formation is supported by analysing the

average number of HBs per solvent molecule at increasing distances from the solute. The distance was measured between the oxygen atom of the solvent and the centre of mass (CoM) of the RS, intermediate, and PS. Since the reaction passes the TS very quickly, the intermediate was chosen for analysis instead.

As the distance from the CoM of the substrate to solvent molecules increases, the number of HBs for each solvent converges to the average bulk value, with two HBs for water molecules and one for methanol. Fig. 4d displays a peak in the number of HB in the water close to the substrate, demonstrating the higher number of HBs in the first solvation shell for the RS (2.13), intermediate (2.43), and PS (2.49) compared to bulk water. Such an increase suggests the organisation of water molecules around the substrate during the reaction. In contrast, no such peak is observed in methanol, indicating that the solvent HB network is not influenced by the presence of substrate. The organisation of solvent molecules should lead to a reduction in free energy, as previously stated. Magsumov et al. demonstrated a linear relationship between the free energy of cavity formation and the volume of the cavity for various solvents through MD simulations utilising classical force fields[71]. Applying their parameters for water and methanol, the change in cavity contribution from RS to post-TS was approximately $-2.6$ kcal mol$^{-1}$ for the *endo* reaction in water, compared to around $-1.1$ kcal mol$^{-1}$ in methanol, further evidencing that the reactions benefit more from the hydrophobic effect in the water than in methanol. These observations thus further support the role of the hydrophobic effect in the acceleration of DA reaction in water.

## Discussion

We have presented a robust and efficient AL strategy that utilises descriptor-based selectors to train MLPs for condensed-phase reactions. We evaluated the impact of three different selectors: *energy*, *similarity*, and *distance* on data efficiency and MLP accuracy using bulk water as a model system. Our findings reveal that descriptor-based selectors outperform the classical *energy* selector in both efficiency and accuracy. This is evident from the smaller number of configurations required with comparable MAD in energies and forces compared with *energy* selector.

We applied this approach to investigate the influence of solvent molecules in the reaction between CP and MVK in water and methanol. The resulting MLPs enable us to efficiently obtain accurate PESs and FESs for these reactions. Consideration of explicit water solvent leads to the formation of a short-lived intermediate, revealing the interplay between hydrogen bond interactions and hydrophobic effects in chemical reactions. The results emphasise the importance of incorporating explicit solvation in the modelling of chemical processes to provide a more realistic representation of experimental conditions.

This work applies the SOAP approach, which is based on 3-body features, for the descriptor-based selectors. Although we demonstrated the advantages of using SOAP-based selectors in automated AL, these selectors may face challenges when applied to highly complex systems due to their geometrical incompleteness. In such cases, descriptors with higher-order features, such as ACE, would serve as a suitable replacement for the SOAP descriptors within the same framework. Another potential limitation arises from the locality assumption used in the SOAP. This means that the descriptor-based selectors may be unable to differentiate between chemical environments that only differ in a long-range region beyond the descriptor cut-off. In such cases, employing a larger cut-off or combination of short-range and long-range descriptors would be beneficial. The use of both injective and concise descriptors, as well as the development of long-range MLPs, are areas currently being explored. In future work, we aim to update the descriptor-based selectors following these advancements.

By harnessing the speed and data efficiency of this framework, we demonstrate the feasibility of accurately modelling chemical processes in explicit solvents using MLPs. We also acknowledge that as the complexity of chemical reactions in solution increases, the identification of multiple potential pathways becomes challenging. To address such challenges, particularly when multiple pathways may exist or when solvent actively participates in the reaction, the integration of enhanced sampling techniques, coupled with chemical intuition within the AL framework, will further enhance the robustness and versatility of our approach. Our approach offers an accurate and efficient means to model reactions in explicit solvents, which will contribute to a deeper understanding of solute-solvent effects and entropy in reactivity.

## Methods

### ACE Training

ACE MLPs were trained using ACE.jl[72] wrapped by pyjulip in *mlp-train* package[43]. Unless specified otherwise, training for ACE MLPs uses hyperparameters listed in SI § S2. All ACE MLP-MD simulations were performed using the Atomic Simulation Environment (ASE) v.3.22.0[73]. package with a timestep of 0.5 fs. Initial velocities were sampled from a Maxwell-Boltzmann distribution. We used the following system-specific ground truth methods: PBE0-D3BJ/def2-SVP[74] for the water system, and ωB97M-D3BJ[75]/def2-TZVP level of theory for the DA reaction. The PBE0-D3BJ method was selected as it provided good agreement with the bulk water structure[76]. The ground truth for the DA reaction was chosen based on a benchmark relative to SCS-MP2[77] (SI § S4). Implicit solvation was accounted for using the Conductor-like Polarizable Continuum Model (CPCM)[78] model. All QM computations were performed in ORCA v. 4.2.1[79] wrapped with autodE[80]. The t-SNE maps of training data sets for bulk water were generated with scikit-learn 1.3.2[81].

### Free energy computations

The 1D free energy profiles in implicit and explicit solvents were computed by umbrella sampling (US). The reaction coordinate was defined as $(r_1 + r_2)/2$, where $r_1$ and $r_2$ are the distance of C-C bonds formed during the reaction (Fig. 4, upper part). The initial configurations were generated by the partial Nudged Elastic Band (pNEB)[82] method with 15 images. Solvent molecules were equilibrated before and during the NEB calculations. The ACE MLP-MD/US was performed with 30 windows for 10 ps per window at 300 K in the NVT ensemble. 15 windows were equally spaced in [1.55, 4.0] Å with $k = 10$ eV Å$^{-2}$, and an additional 15 were equally spaced over [1.7, 2.5] Å with $k = 20$ eV Å$^{-2}$, except for the *Exo* reaction in explicit water, for which $k = 13$ eV Å$^{-2}$ and $k = 30$ eV Å$^{-2}$ were applied for the wide- and narrow-range, respectively, to ensure overlap among the windows.

### Trajectories with explicit solvents

The dynamic properties of the DA reaction in an explicit solvent were investigated by propagating downhill and uphill trajectories. Downhill trajectories were propagated from the TS, while uphill trajectories were initiated from RS.

Initial configurations for the downhill trajectories were obtained using a method inspired by the solvent-perturbed transition state (SPTS) sampling scheme proposed by Houk et al.[70] TS frames for reactions in explicit solvents were initially selected from ACE MLP-MD/US. These frames were equilibrated for 20 ps in the NVT ensemble while fixing the solute geometry. Snapshots were collected every 5 ps from resulting 20 ps trajectories. In these snapshots, TSs were optimised using the Broyden-Fletcher-Goldfarb-Shanno (BFGS)[83-86] method, implemented in ASE, with fixed solvent molecules and harmonic potentials to maintain the TS distance of the forming bonds, obtaining optimised TSs reflecting solvent environments. Downhill dynamics were propagated from optimised TSs until the products or reactants were formed within a maximum simulation time of 2 ps. Product formation was considered when both $r_1$ and $r_2$ distances were

below 1.6 Å, whereas the reactant was considered formed if the forming bond distances exceeded 3.0 Å.

Starting points for uphill trajectory were obtained by independent ACE-MD simulations of reactants at 300 K with $r_1$ and $r_2$ constrained by harmonic potentials at 3.0 ~ 5.0 Å, respectively. The uphill trajectories were propagated with a harmonic potential (spring constant of 0.4 eV Å$^{-2}$) to overcome the energy barriers. Trajectories were propagated until products formed or the simulation time reached 3 ps.

### Reporting summary

Further information on research design is available in the Nature Portfolio Reporting Summary linked to this article.

## Data availability

The training datasets used in this study, along with the initial and final configurations of downhill and uphill trajectories for the DA reactions, have been deposited in the Oxford Research Archive (ORA) under accession code https://doi.org/10.5287/ora-zbjyr1ybk. Source data are provided with this paper.

## Code availability

*mlp-train* package is available at https://github.com/duartegroup/mlp-train[87].

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

## Acknowledgements

The authors thank Dr T. Piskorz, Dr A. Sterling and Z. Zhu for insightful discussions. They also acknowledge Dr H. Chan, Dr Z. Bo, Dr B. Lee, N. Frank, V. Vitartas, and Dr T. Johnston-Wood for their feedback on the manuscript and codes. H.Z. thanks the EPSRC Centre for Doctoral Training in Theory and Modelling in Chemical Sciences (EP/L015722/1). V.J. acknowledges the funding from the Swiss National Science Foundation (SNSF, Postdoc. Mobility fellowship, grant no. 210737). This work used the University of Oxford Advanced Research Computing (ARC) facility and the Cirrus UK National Tier-2 HPC Service at EPCC, funded by the University of Edinburgh and EPSRC (EP/P020267/1).

## Author contributions

H.Z. and F.D. conceptualised the study. H.Z. carried out the calculations. All authors participated in data analyses and writing of the manuscript. H.Z. and V.J. wrote the first draft. F.D. supervised the study.

## Competing interests

The authors declare no competing interests.
