## [Peer Review File · Nature Communications]

Modeling Chemical Processes in Explicit Solvents with Machine Learning PotentialsREVIEWER COMMENTS

Reviewer #1 (Remarks to the Author):

This paper reports a novel approach for generating machine learning potentials (MLPs) to model chemical processes in solution. The strategy proposed relies on active learning, where new configurations are added based on their position in the molecular-descriptors (i.e. SOAP) space. This feature allows to span more efficiently the chemical and conformational space at low computational cost. Two descriptors were tested first by investigating a simple water box, to determine the quality of the potential and data efficiency. Then the strategy has been applied to study the Diels-Alder (DA) reaction between cyclopentadiene and methyl vinyl ketone in two different solvents: water and methanol. The manuscript is well written the issue well defined and both methods and results are well explained.

However, there are some concerns that need to be addressed prior to publication:

- The first major one regards the ability of the strategy adopted to sample the relevant chemical and conformational space. The active learning strategy adopted is based on short MD simulations (max 5 ps) using the first version of the trained MLP. These are started from configurations already present in the training set containing reactants, products, and transition state. As a consequence, I assume the MLP to be very good at modelling the three states. However, I would like to see if good sampling has been obtained also along the reaction path connecting the three.
- In addition, I would like to ask the authors to comment on the ability of the present strategy to model more complex system, where different paths may be activated by the presence of the solvent molecules. Would the same strategy be still effective? Or would it be necessary to include enhanced sampling techniques in the active learning strategy to explore effectively the relevant chemical space?
- The second concern regards the validation of the ML potential trained to model the DA reaction. This is based on a MLP-MD simulation of a box containing the substrate and 55 water molecules of 500 fs. The limited time makes me wonder if the MLP is actually able to reproduce the rearrangement of the H-bonds in the solvent around the substrate along the reaction path and correctly account for their contribution to the reactive process. Therefore, in my opinion, longer simulations need to be performed to ascertain the ability of the potential to correctly model the dynamics of the solvent + substrate system.
- In addition, it is not specified whether the MLP-MD simulations used to validate the potential are performed in the NVT or NVE ensemble. Since entropy is a major player in this reaction, I think it is important to determine if the thermal fluctuations of the solvent + substrate system are well captured by the potential with equilibrated NVT simulations.

Reviewer #2 (Remarks to the Author):

Review of Zhang et al.

This manuscript presents machine learned potential energy surfaces for reactions in solution. The main theme is the selection of reference structures based on different information: "energy", "similarity" and "distance" from ensembles of reference structures. Primarily, the trained models are then applied to a typical Diels-Alder reaction from running collections of short MD simulations. Reference data is determined at the DFT level and some validation with geometries from MP2-SCS calculations were carried out. Nevertheless, the level of theory is rather moderate and not expected to be quantitative for absolute reaction free energies. Overall, the work was carried out carefully but the findings are not entirely new and surprising.

Detailed comments:

1. The accuracy of the models appears to vary between 0.35 and 7.6

kcal/mol - see Figures in SI. For the specific case of explicit water (Figure S9) RMSE ranges from 1.2 to 7.6 kcal/mol. This is not sufficiently accurate for the claims on p. 15, namely that the agreement with experiment is favourable and improves with explicit water. Furthermore, although tested on geometries from MP2-SCS calculations (and apparently not on energies), all methods used in the present work are far from truly quantitative reference methods such as CCSD(T). Is the "agreement with experiment" not rather fortuitous?

2. What are the sizes of the training, test and validation sets?

3. Are downhill trajectories really suitable for investigating the (a)synchronicity of a reaction? How much does omitting the transit from reactant to the TS affect the dynamics and therefore the (a)synchronicity? Note that the question of (a)synchronicity for DA reactions (in gas phase - i.e. comparable to simulations in "implicit solvent" here) has also been considered for 2,3-dibromo-1,3-butadiene and maleic anhydride, see Rivero et al. (JCP 2019; Mol. Phys. 2020).

4. Similar effects as to transient formation of the zwitterion were also reported for double proton transfer in formic acid in solution using machine learned PES, see Töpfer et al., PCCP 2022.

5. To put the energetics discussed in the main MS and in the SI in perspective, using identical units throughout will be helpful.

6. The validation in the SI for energies and forces appears to have been carried out for idealized scenarios ("gas-phase TS geometry immersed in 54 water molecules, which was equilibrated by 1 ps MD simulation with fixed TS before propagating"). Would it not be more natural to just pick samples from equilibrium or the US trajectories?

In summary, while most of the work was carried out carefully, the novelty appears to be in selecting the reference structures for training, which is rather technical. Also, it is unclear, how robust the trained models in particular for the reaction in water really are, given the large MAE.

Reviewer #3 (Remarks to the Author):

This work proposes an active learning (AL) approach following a descriptor-based selection for the new configurations to update a machine learning potential (MLP). This approach is used to model and analyze the Diels-Alder (DA) reaction between cyclopentadiene (CP) and methyl vinyl ketone (MVK) in explicit solvents (water and methanol). The authors demonstrate that the proposed descriptor-based selectors for AL outperform typical energy-based selectors both in terms of efficiency and accuracy. Moreover, the obtained MLPs using this AL procedure are robust and help the authors in obtaining relevant thermodynamic properties of the reaction with both solvents, matching experimental results published in the literature.

From my point of view, this paper represents a very nice piece of scientific work and, as the authors say, it indeed represents a clear step forward in the development of MLPs that can handle an oftentimes tricky task: modelling solvation. With this being said, I am happy to recommend this work for publication in Nature Communications after the authors address some minor comments that can polish what it is already discussed in their work:

1) When the authors discuss the state-of-the-art of MLPs for solvation in the Introduction (specifically in the first paragraph in page 3), the authors miss to refer to relevant reference, for instance, <https://pubs.aip.org/aip/jcp/article/155/8/084101/1018119/Machine-learning-implicit-solvation-for-molecular>. It would be good if the authors could look again for references in case there are other previous works that might be useful to highlight even more the breakthrough that the present work represents.

2) At the end of page 4 the authors say: "In contrast to the metrics described above, these selectors use molecular descriptors, such as Smooth Overlap of Atomic Positions (SOAP) or ACE. to assess whether the training set accurately represent the chemical space of interest". This sentence makes the reader to think that the authors will use both descriptors to construct their selector, but later in the text it seems that only SOAP was used for such task. Could the authors please clarify this point and maybe rewrite part of the sentence to avoid confusions?

3) The authors proved in their work that the descriptor-based selectors outperform the typical energy-based ones. However, I would like the authors to spend a few sentences to discuss the possible drawbacks that can become relevant once their approach is used in larger (and possibly more flexible) systems. Currently, few examples come to my mind:

i) Descriptors such as SOAP, and even ACE, neglect all (or at least some) long-range features. For larger systems, it could be the case that molecular geometries share very similar local environments, while their long-range features are different. In such a case, the descriptor-based selector could indicate that these configurations are similar when they are not.

ii) Even for medium-sized systems, the authors of the original SOAP's paper themselves have proved that the SOAP descriptor can have problems with its "uniqueness" (i.e., two geometries that are different share the exact same SOAP descriptor).

This could be a great asset for the Conclusions/perspectives of the work.

4) Figure 1 misses the letter "a" in Top Figure.

5) In page 7, in the last paragraph of the "Workflow" subsection, the authors mention that the time of the simulation is set to $(n^3 + 2)$ fs, but the timestep is missing. I consider that it is relevant in order to know how many configurations are sampled in each run.

6) In page 8, the authors mention that "Selecting an appropriate threshold is key as too low values (e.g., similarity below 0.9) can result in the selection of non-physical structures that fail to converge in the self-consistent field (SCF) computations, while too high values (e.g., 1) do not provide any additional information.". With respect this sentence:

i) Does this mean that you only select configurations with similarity values between 0.9 and 1?

ii) Do you obtain non-physical structures from the MDs run with your MLP? If so, what do you do within your framework to deal with this problem: do you select the last "physical" structure available?

7) The authors use the Euclidean distance as metric in the calculation of the local outlier factor (LOF). It is known that in high-dimensional spaces (such as the one spawned by SOAP vectors) the Euclidean metric is not as reliable as in lower-dimensional spaces. Have the authors explored the use of other metrics in the calculation of the LOF? Maybe for the example proposed in the present work there would not be differences, but for other (larger) systems it could be relevant. Do the authors consider that this could be an issue for extending the application of their approach to other systems of interest?

8) In the previous-to-last paragraph of page 9, the authors mention: "we chose the threshold to encompass 20% of the outliers present in the training data". This is not clear to me. What does this mean? How do they define/select the outliers in the training?

9) In terms of data availability, it would be good if the authors also make their datasets available.

Reviewer #4 (Remarks to the Author):

Summary:

In this article, the authors present two original methods to accelerate the training of reactive machine learning potentials (MLPs) through active learning (AL) selection strategies generally referred to as "descriptor-based selectors." The first approach, termed similarity, consists of computing a similarity metric between new structures and previously observed ones. New structures are only added to the training set if their similarity across all previous observations is lower than a user-selected threshold. The second approach, termed distance, consists of computing the local outlier factor (LOF) for new structures given the previously observed ones. Then, the new data points are ranked by their LOF and the top 20% is added to the training set. Both methods follow an AL workflow, where the model is fitted again on the new training set and new data is collected until no new conformations can be added to the data set.

The researchers benchmarked their methods in comparison to an energy-based selector in water box simulations. They found that both similarity and distance achieved mean absolute deviation (MAD) errors below 1 kcal/mol and 2 kcal Å/mol (for energy and forces, respectively) while requiring only a small fraction of the data needed by the baseline. Moreover, since similarity and distance concern only geometrical aspects of the data distribution, QM calculations are not necessary to compute the energy error, which saves time compared to the baseline. Although Figures S1-S3 suggest that distance was the best method in terms of MAD error, the authors claim that this approach requires too many initial data points and proceed to study a Diels-Alder (DA) reaction system in explicit solvent using similarity-trained MLPs only.

The authors deploy a wide range of computational techniques to characterize the DA reaction of choice and conclude that the MLP trained under the similarity selector was helpful in obtaining accurate potential and free energy surfaces.

To my knowledge, this work is the first to introduce an AL loop that combines LOF and similarity selectors with molecular descriptors (SOAP and ACE). Even though all of these methods are previously reported in the literature, the combination is original and is likely to be of interest to the field of ML-based reactive potentials. However, I encountered certain points where the decisions made by the authors are not clearly supported with evidence or the results are not presented in light of broader context. For this reason, I recommend major revisions before the work is published.

Comments:

Since my area of expertise lies in machine learning applied to molecular dynamics simulations of biological systems, I will restrict my comments to the methodological advances proposed by the authors rather than the chemical insight offered about the DA reaction.

1. The authors show enough evidence to prove that their methods are more efficient compared to the energy baseline using bulk water as a simple study case. The authors also claim in the Conclusion that the MLPs obtained with their selectors are more accurate than those obtained with energy. I agree with the efficiency claim because the number of conformations required by similarity and distance was much lower compared to the baseline, but the energy MAD error for similarity was higher than that of energy (Figures S1-S2). Since similarity was used throughout the study, the authors should note in the Conclusion that the energy MAD for energy was lower than that for similarity and then explain if the difference is negligible or not.

Maybe another way to run this comparison, if possible, would be to restrict the total number of conformations that energy can train on to match the final number of conformations selected by similarity (40) and then compare the MADs. I would expect the MADs for energy to be higher because the conformation set would be less diverse. I think this comparison would showcase the advantage of similarity, which would hypothetically be more accurate than energy with an equal number of training samples (but not in a more general sense).

2. About the decision not to continue using distance, did you run any tests with the DA system, either in implicit or explicit solvent, before discarding this option? There could be a more thorough explanation of how much more initial data distance requires in comparison to similarity. The MLP used to study the reaction is the one obtained at the end of the entire training process, so I don't understand why the lack of initial data would be an impediment. Maybe distance performs worse during the early iterations of the AL cycle due to the initial data requirement, but does this tell us anything about the performance of the final MLP?

3. The ACE MLP used to study the DA reaction in explicit water was obtained by training it with a total of 600 conformations belonging to four different subsets. How were these subsets obtained? My understanding is that the conformations were chosen using the similarity selector, but maybe I am misinterpreting how you set up the experiment. This could be clarified in the text or the SI.

4. The authors claim that their selection strategies save time in comparison to energy. I am convinced that this is the case. However, as the authors explain, there are two reasons why their strategies save time: fewer conformations need to be collected before the MLP converges to <1 kcal/mol MAD, and QM calculations are avoided. What is the relative contribution of each factor to the time saved?

5. Since the AL loop ends when no more conformations are selected, it is possible for the training

to stop before the conformational space has been sampled if the user chooses bad hyperparameters. Are there any suggestions for hyperparameter selection? Are the values that you chose for this study good defaults in general? If the users have to do exhaustive hyperparameter tuning, this might make the usage of your selection strategies more difficult.

6. There seems to be no discussion about the relative burden of training the MLP vs. actually sampling the reaction pathway of a reactive system in explicit solvent. The authors claim that their workflow can be used generally to study reactions in explicit solvents, but I am unsure if the "workflow" refers simply to the AL algorithms or to the entire process of studying the reaction in explicit solvent. If the latter is the case, then I think that there are too many system-specific choices (e.g., umbrella sampling reaction coordinate) for the workflow to be widely applicable. If the former is the case, then I suggest that the authors discuss what is the advantage of using the selector-based training in the broader context of characterizing an arbitrary reaction in explicit solvent. It is possible that the actual bottleneck for a future study that applies your workflow is the sampling of the reaction pathway rather than the convergence of the MLP, and your AL algorithms only accelerate the training part.

7. A minor observation: K is called the "similarity matrix" of a point, but it seems the dimensions are $1 \times N$ (where N = number of conformations already in the training set). If this is correct, it might be clearer to say K is a vector of length N .

Reviewer #5 (Remarks to the Author):

Modeling Chemical Processes in Explicit Solvents with Machine Learning Potentials' (NCOMMS-23-36219)

Reviewer #1 (Remarks to the Author):

1. The first major one regards the ability of the strategy adopted to sample the relevant chemical and conformational space. The active learning strategy adopted is based on short MD simulations (max 5 ps) using the first version of the trained MLP. These are started from configurations already present in the training set containing reactants, products, and transition state. As a consequence, I assume the MLP to be very good at modelling the three states. However, I would like to see if good sampling has been obtained also along the reaction path connecting the three.

We appreciate the reviewer's insightful comments. To ensure we effectively sample along the reaction path, our training approach involves the generation of four subsets, each targeting different interactions in the system to describe intrinsic reactivity, various solute-solvent interactions, and solvent-solvent interactions. For each subset, excluding the one comprising only solvent, molecular dynamics (MD) simulations (MLP-MD) are initiated from the transition state (TS) using the first version of the MLP. These simulations are propagated both forward (towards products), and backwards (towards reactants) for a duration of $(2 + n^3)$ fs, where 'n' is an integer iteratively increased from zero until the dynamics in the active learning (AL) loop reach the maximum time of 5 ps, at which point no new data are found by the selector. As shown below, this simulation time was found to be sufficient to sample the chemical and conformational space of the DA reaction. We acknowledge that for more complex systems, an adjustment in simulation time may be necessary - especially for larger and flexible systems. In such cases, confirmation changes may lead to the sampling of several minima, requiring longer times to reach the global minima in either direction.

We have now included a new figure in the SI (**Fig. R1, Fig. S10**), illustrating the distribution of configurations in the training dataset for the *endo* reaction in explicit water in terms of the forming bond distances, r_1 and r_2 . The white circle in **Fig. R1(a)** represents the starting point for AL, corresponding to the TS obtained in the gas phase. The white dashed line depicts the reaction path in explicit solvent connecting reactant state (RS), TS, and product state (PS) obtained from partial nudged elastic band (NEB) calculation for the reaction solvated with 200 water molecules. The NEB was run using the final MLPs with optimised RS/PS in water used as the initial points. The orange circle corresponds to the highest energy point from the NEB profile. While this point does not overlap with the TS in the gas phase, the small difference observed agrees with the expected variation between the gas phase and explicit water TS. This demonstrates that even though our AL starts from a gas-phase TS, it learns the relevant chemical space for the condensed-phase reaction. The counterplot in **Fig. R1(a)** visually represents the "closeness" of points on the potential energy surface (PES) to those in the training set. Dark regions denote the closest points to the training, while orange regions indicate areas less well represented. This analysis confirms that the MLPs are trained using the structures covering the whole reaction path. **Fig. R1(b)** illustrates the location of each training data point in the PES, coloured by corresponding subsets. Details on each subset are provided below in **Table R1 (Table S6)**. Note that some data points in **Fig. R1(b)** corresponding to the RS region are outside the PES presented in **Fig R1(a)**, providing more details on structures where reactants are further apart. This figure further demonstrates that our AL method adequately samples the full reaction profile.

Figure R1 (S10). Closeness and locations of training data for the endo reaction in water. (a) The relative distance between each point in the training data and the reaction PES generated by the 2-D scan, measured by $\min(r_k-r)$, which represents the minimum distance, averaged between r_1 and r_2 , between the point on the surface and all points in the training dataset (See SI §S6.2 for details). The darkest regions ($\min(r_k-r)=0$) correspond to data in the training set, while pink/orange regions indicate areas less well-represented. The white circle corresponds to the gas phase TS, used as a starting point for AL. The white dashed line depicts the reaction path in explicit 200 water molecules, with the highest energy highlighted in orange. (b) The location of training data is colour-coded based on the origin of the data point. The region corresponding to the PES shown in (a) is enclosed by dashed lines.

Furthermore, we have also elaborated on how the training data are generated for each subset and how they are subsequently combined to obtain the final MLP, as some of the confusion may arise from the original, more concise version describing our training strategy. We emphasize the importance of this iterative AL process to ensure the inclusion of all relevant structures along the reaction path. This description has been included in the SI alongside Fig. R2 describing the training strategy in more detail (Fig. S9 in SI §S6.1), as presented below for easy reference.

Detailed Description of Training Strategy:

The overall training set consists of four subsets, each aimed at describing different interactions in the reactive system (Table S6). **Subset 1** corresponds to the structures generated from the transition state (TS) of the system in vacuum (CP + MVK) and provides information about intramolecular interactions and intrinsic reactivity (as shown in Fig. 3(a) in the main text). **Subset 2** consists of the structures generated from the same gas-phase TS, this time microsolvated with either two water molecules or one methanol molecule, depending on the solvent used. The solvent molecules were randomly placed around the solute carbonyl group, constrained to form hydrogen bonds. **Subset 3** corresponds to the gas-phase TS structure now fully solvated with 33 explicit water or methanol molecules. Subsets 2 and 3 aim to describe various solute-solvent interactions occurring during the reaction. Finally, **subset 4** contains only solvent molecules (27 water or methanol molecules), providing information about solvent-solvent interactions in bulk solvent.

Figure R2 (S9). Training strategy implemented in this study. a) Schematic representation of the active learning strategy. The grey box illustrates the detailed procedure of MLP-MD sampling and selector evaluation. b) Generation of training data for the endo reaction in water, showcasing representative initial configurations for each subset.

All subsets were generated independently using the AL scheme (Fig. S9(a)) with the structure used to initiate the training listed in Table S6, as outlined in Fig. S9(b). Since one structure is not enough to train the first version of the MLP, we generated 10 initial configurations as follows: For subsets 1 and 2, initial configurations were generated by the random displacement of atomic positions of the gas-phase TS structure. Since the ACE potential requires information on the box size for all structures in the training set, subsets 1 and 2 were assigned a cubic box of 100 Å. For subsets 3 and 4, the 10 initial configurations were generated by randomly placing the solvent molecules in a cubic box of 11 and 9.32 Å, respectively, to ensure the water density was reproduced. For methanol, the box was adjusted according to its density of 792 kg/m³ at room temperature. To train the ACE potential and propagate dynamics in the AL, the size of the boxes was then set to 100 Å without changing the positions of the molecules to create a solvated cluster in the gas phase.

For all subsets, the 10 initial configurations were used to train the first version of MLP. We trained one MLP for each of the subsets and used it to propagate short MLP-MD forward (product) or backward (reactants) from the TS and continue AL. The direction was assigned randomly based on the initial velocities from a Maxwell-Boltzmann distribution at 300 K. In all cases, MLP-MD was propagated for $(2 + n^3)$ fs, using a time step of 0.5 fs, with n (integer) initiated at 0. The chosen selector was then applied to evaluate the last frame of the trajectory to determine if the structures would be added to the existing training set. If the selector did not find any new data, the value of n increased until it reached the maximum time of 5 ps. This approach ensures that the full reaction profile is sampled in the AL and that all relevant structures along the path are included in the data set. As shown in Fig. S10, this simulation time was sufficient to sample the chemical and conformational space of the reaction. We acknowledge that for more complex systems, an adjustment in simulation time may be necessary; in particular for larger systems, where conformation changes lead to several minima requiring longer times to reach the global minima in either direction.

Table R1(S6). Number of configurations obtained after AL and the final training set selected for different reactions.

Subset	Initial structure for training	Reactions	# Configs.	
			After AL	Selected
1	gas-phase TS of CP+ MVK (TS _{CP+MVK})	endo in water	215	150 for each
		exo in water	229	
		endo in methanol	215	
		exo in methanol	229	
2	TS _{CP+MVK} + 2 waters / 1 methanol molecule(s)	endo in water	230	150 for each
		exo in water	258	
		endo in methanol	215	
		exo in methanol	235	
3	TS _{CP+MVK} + 33 solvent molecules	endo in water	260	250 for each
		exo in water	259	
		endo in methanol	252	
		exo in methanol	255	
4	pure solvent, either 27 water or methanol molecules	endo in water	52	50 for each
		exo in water	52	
		endo in methanol	81	
		exo in methanol	81	

After generating these four datasets by AL, they are combined by randomly selecting a specific number of configurations listed in **Table S6** to train the final MLP. Each of the datasets obtained from the AL was reduced by 1 – 42 % to reduce the memory requirements for training the MLP on all data points, as the ACE descriptor scales as S^v (where S is a number of elements and v represents the correlation number) for each data point. Data points from Subsets 1 and 2, corresponding to the reaction in the gas phase and microsolvated environment, were reduced by 30 – 35 % and 30 – 42 %, respectively. This decision is driven by the fact that the structural information provided by these subsets overlaps significantly. Data in subset 3 was reduced by only 1 – 4 % to ensure a proper description of solvent–substrate interactions. These reduced datasets were then combined, resulting in the final 600 configurations for training for each reaction. The combination of gas-phase and microsolvated data ensures that the training set contains information about the relevant states (RS, TS, PS) configurations along the reaction paths and relevant interactions for each of them. The use of cluster subsets in the training makes the process much more efficient compared to computationally more expensive periodic Density Functional Theory (DFT) calculations. Moreover, this approach enables the use of higher-level theory calculations, both in terms of method and basis sets. This is a limitation in periodic *ab initio* MD (AIMD) calculations, which are typically restricted to GGA or, at most, hybrid GGA functionals with specialized plane-wave basis sets.

We have also modified the paragraph in the main text, section “Training Strategy – DA Reaction of CP and MVK in Explicit Water”, page 6.

For the reaction in explicit solvents, the training set consisted of four subsets, each aimed at describing different types of interactions in the system. Subset 1 corresponds to the substrate complex (CP + MVK) and provides information about intramolecular interactions and intrinsic reactivity (**Fig. 3(a)**). Subsets 2 and 3 consist of the substrate with 2 and 33 water molecules, respectively, and aim at

describing various solute-solvent interactions. Finally, subset 4 contains only water molecules, providing information about solvent-solvent interactions in bulk solvent. All subsets were generated independently using the AL scheme initiated from the transition state structure, except the pure water subset, which started from a random water configuration. This approach ensured that the training set contained reactants, products and connecting reaction paths for all studied environments. The combination of these sub-training sets yielded 600 training points, which we used to train the final ACE MLP. See **SI §S6.1** for further details.

2. In addition, I would like to ask the authors to comment on the ability of the present strategy to model more complex system, where different paths may be activated by the presence of the solvent molecules. Would the same strategy be still effective? Or would it be necessary to include enhanced sampling techniques in the active learning strategy to explore effectively the relevant chemical space?

Our active learning approach is designed to be flexible in handling systems of increasing complexity. The efficiency of the approach relies on two essential components: the sampling of the process of interest and the selection of relevant configurations for training. The novelty of our work lies in the latter, streamlining the selection process by i) employing chemically meaningful initial configurations that enable the model to learn relevant interactions and ii) using computationally inexpensive descriptors based on similarity measurements or outlier detection to select new configurations. This eliminates the need for a large number of training configurations and QM references for selection during the training process.

In our study, we intentionally chose a reaction with known pathways to focus on evaluating the performance of the selectors. The studied reaction can proceed via an *endo* or *exo* TS, and we have chosen to train separate MLPs for each pathway. However, we acknowledge that in scenarios where the pathways are unknown, or when solvent may participate in the reaction, the sampling of the relevant processes becomes more challenging. While unbiased long-time MD simulations at elevated temperatures could be used to “discover” relevant processes, it would come at a significantly higher cost in terms of number of configurations required and training time. To address these challenges, we are actively working on incorporating enhanced sampling techniques into our AL framework, as already presented in the *mlp-train* package (<https://github.com/duartegroup/mlp-train>; publication in preparation). This is crucial for overcoming high-energy barriers without sampling unphysical regions, a limitation associated with running unbiased MD at high temperatures. The combination of chemical intuition, allowing users to suggest potentially relevant states for exploration, alongside the use of automated pathway search tools such as *autodE*, and enhanced sampling methods will enable a more comprehensive exploration of complex systems, offering a cost-effective alternative to current AIMD-based approaches. Crucially, the selection schemes compared here are applicable regardless of the sampling methods employed or the complexity of the system, showcasing its potential for broader applications.

To address the reviewer’s concerns and clarify this point, we have modified the discussion section in the main text, page 10:

We also acknowledge that as the complexity of chemical reactions in solution increases, the identification of multiple potential pathways becomes challenging. To address such challenges, particularly when multiple pathways may exist or when solvent actively participates in the reaction, the integration of enhanced sampling techniques, coupled with chemical intuition within the AL framework, will further enhance the robustness and versatility of our approach. Our approach offers an accurate and efficient means to model processes in explicit solvents, paving the way for more thorough investigations of complex processes in solution. This will contribute to a deeper understanding of the effects of solute-solvent interactions and entropy in reactivity.

3. The second concern regards the validation of the ML potential trained to model the DA reaction. This is based on a MLP-MD simulation of a box containing the substrate and 55 water molecules of 500 fs. The limited time makes me wonder if the MLP is actually able to reproduce the rearrangement of the H-bonds in the solvent around the substrate along the reaction path and correctly account for their contribution to the reactive process. Therefore, in my opinion, longer simulations need to be performed to ascertain the ability of the potential to correctly model the dynamics of the solvent + substrate system.

To ensure our MLP training time was sufficient, we conducted additional validation dynamics over 3 ps with a timestep of 0.5 fs for the *endo* reaction in both explicit water and methanol. This extended simulation time provides a longer time window for H-bond rearrangement and solvent reorganization in the solution and, therefore, a more comprehensive assessment of the MLP's reliability in longer-time dynamics. Comparison of energies and forces (Fig. R3, now included in the SI as Fig. S13 and S17 for water and methanol, respectively) show that over the 3ps simulations, MAD in energy slightly increased from 0.7 meV/atom to 0.91 meV/atom for water and 0.3 meV/atom to 0.73 meV/atom for methanol. MAD in forces stayed comparable in both validations, *i.e.*, 59 meV/Å and 63 meV/Å for water, and 54 meV/Å and 58 meV/Å for methanol. The magnitude of the errors in both cases is in line with the existing state-of-the-art approaches, as discussed in more detail in reply 1 to the comments of reviewer 2 (*vide infra*).

Figure R3 (S13 and S17). Performance of the ACE MLP for the *endo* reaction of CP and MVK in water and methanol. Comparisons of ground-truth (ω B97M-D3BJ/def2-TZVP) and predicted (ACE MLP) energies and forces over 3-ps ACE MLP-MD trajectories started from TS (obtained in the gas phase) solvated in 55 water or 40 methanol molecules, which was equilibrated by 1 ps MD simulation with fixed TS before propagating (300 K, time step = 0.5 fs)

4. In addition, it is not specified whether the MLP-MD simulations used to validate the potential are performed in the NVT or NVE ensemble. Since entropy is a major player in this reaction, I think it is important to determine if the thermal fluctuations of the solvent + substrate system are well captured by the potential with equilibrated NVT simulations.

We performed different validation dynamics to assess the accuracy and stability of the potentials and to sample the thermal fluctuations of all systems. The validation dynamics for bulk water presented in **SI §S3.1** in the main text were performed both in NVE and NVT ensembles. NVE MD was performed to evaluate the stability and energy conservation of the system under periodic boundary conditions, using longer dynamics than those employed in the training phase (50 ps vs. 5 ps in AL). On the other hand, NVT dynamics provided information on the accuracy of MLPs for the cluster system at 300 K. For the DA reaction, the MLP-MD simulations propagated during AL and validation dynamics were exclusively performed in the NVT ensemble at 300 K, utilizing the Langevin thermostat. In both systems, the MLPs show sufficient stability and accuracy, as depicted by energy and forces MAD in **Fig. S1-S3, S11-S13, and S15-S17** in the SI. Since NVT MLP-MD simulations are used to validate the performance of MLPs, we are confident that our models accurately capture the thermal fluctuations. To avoid misleading descriptions, we have now clarified the text in the respective sections in the SI as follows:

To assess the accuracy of ACE MLPs for Diels-Alder reactions, 500 fs MLP MD simulations with a timestep of 0.5 fs were conducted in an NVT ensemble at 300 K using a Langevin thermostat. This was followed by a point-to-point comparison of energies and forces between the ACE MLPs and the ground truth method for the structures obtained from the trajectories.

Reviewer #2 (Remarks to the Author):

1. The accuracy of the models appears to vary between 0.35 and 7.6 kcal/mol - see Figures in SI. For the specific case of explicit water (Figure S9) RMSE ranges from 1.2 to 7.6 kcal/mol. This is not sufficiently accurate for the claims on p. 15, namely that the agreement with experiment is favourable and improves with explicit water.

When discussing errors in the trained model, it is important to consider both the model's ability to reproduce the training data and its generalization to unseen data. Moreover, reporting absolute errors (like MAD) across systems of different sizes can be misleading as the magnitude of the potential energy value increases with the number of atoms and the comparison does not reflect the relative errors. Thus, it is important to employ metrics that facilitate a fair comparison across systems of different sizes. To maintain consistency in MAD values across systems of varying sizes, we present all errors in meV/atom and meV/Å for energy and force errors, respectively, in **Tables R2 and R3**, now also part of the **SI §S6.4**. By expressing energy errors per atom, the models' performance in both energy and force remains unaffected by the increasing size of the system.

In the original manuscript, we compared MAD for two systems with compositions different than the training data. Here, we extend the validation to systems with the same composition as the training data, referred to as training-like systems. All data used for testing were generated separately from the training to prevent data leakage and unforeseen extrapolation errors. Each of the four final MLPs (i.e., one for *endo* and *exo* reaction in water and methanol solvents) was evaluated over 4 sets of structures containing 101 structures each, generated by independent 500-fs MLP MD trajectories.

In the table below, we summarise MAD for energy and forces across all the tested systems, *i.e.*, the newly generated training-like systems and previously reported testing system. The training-like systems share the same number of molecules as the corresponding subsets described above (**Table R1**). The test systems include more solvent molecules than the training data. For instance, the training-like set corresponding to subset 2 in **Table R1** represents the gas-phase TS bound to two (one) water (methanol) molecules. In contrast, **test sets 1 and 2** include one more solvent molecule, *i.e.*, three

(two) solvent molecules for water and methanol, respectively (entries 1 and 2 in **Table R2**). Similarly, subset 3 in the training-like set consists of the TS immersed in 33 water or methanol molecules, while **test sets 3 and 4** have 55 and 40 solvent molecules, respectively (entries 3 and 4 in **Table R2**). For sets 1 and 2 in **Tables R2 and R3**, the simulations were initiated from structures with randomly placed solvent molecules. The resulting dynamics thus included the equilibration of solvent molecules, as evident from the rapid decrease in energy shown in **Figs. S11, S12a, S15a and S16a**. The energy and force errors in these systems range from 1.27 meV/atom to 3.56 meV/atom and 62 meV/Å to 116 meV/Å, respectively. In the second case, *i.e.*, sets 3 and 4 in **Tables R2 and R3**, solvent molecules were equilibrated using MLP-MD before running the validation MD. The energy and force errors for water and methanol for solvated systems vary from 0.3 meV/atom to 1.8 meV/atom and 52 meV/Å to 62 meV/Å. These validations indicate that adding more solvent molecules does not deteriorate the accuracy and the potentials have sufficient transferability across systems with different sizes. The large errors for testing in entries 1 and 2 in **Tables R2 and R3** are likely due to distorted structures in the validation trajectory, as these MDs were initiated from randomly placed solvent molecules. While testing MLPs on trajectories starting from randomly placed molecules may seem counterintuitive, this approach challenges the performance of MLPs in modelling distorted structures and out-of-equilibrium geometries.

It is also useful to compare the performance of the MLPs reported here to state-of-the-art potentials developed for reactive systems of similar complexity. For instance, the DeepMD NN potential trained for urea decomposition in water (*Catal. Today* 387 (2022) 143–149) has reported MAD 6.05 kJ/mol (1.44 kcal/mol) for the energies and 3.54 kJ/mol/Å (0.85 kcal/mol/Å) for the forces on the test set with 14,536 training data points. For a system with the same composition as in the training data of 1 urea molecule and 34 water molecules (110 atoms), these MADs correspond to **0.57 meV/atom** and **36.69 meV/Å**. In our case, the MAD for a system of similar size (*i.e.*, a training-like system of reactions in explicit water) is **0.48 meV/atom** (*i.e.*, 1.34 kcal/mol, 5.60 kJ/mol) for energies and **58 meV/Å** (*i.e.*, 1.33 kcal/mol/Å, 5.59 kJ/mol/Å) for forces for the *endo* reaction in water, and **0.97 meV/atom** (*i.e.*, 2.71 kcal/mol, 11.32 kJ/mol) for energies and **58 meV/Å** (*i.e.*, 1.33 kcal/mol/Å, 5.59 kJ/mol/Å) for forces for the *exo* reaction in water. These errors showcase comparable accuracy to the DeepMD NN potential using an order of magnitude fewer training data.

The additional section on the summary of MLPs' performance in SI is as follows:

In **SI §S6.2-6.3**, we compared MAD for two systems with different structures than the training data. Here, we extend the validation to systems with the same composition as the training set, referred to as training-like systems. All data used for testing were generated separately from the training to prevent data leakage and unforeseen extrapolation errors. Each of the four final MLPs (*i.e.*, one for *endo* and *exo* reaction in water and methanol solvents) was evaluated over 4 sets of structures containing 101 structures each, generated by independent 500-fs MLP MD trajectories.

In the table below, we summarise MAD for energy and forces across all the tested systems, *i.e.*, the previously reported testing system and newly generated training-like systems. For sets 1 and 2 in **Tables S7 and S8**, the simulations were initiated from structures with randomly placed solvent molecules. The resulting dynamics thus included the equilibration of solvent molecules, as evident from the rapid decrease in energy shown in **Figs. S11, S12a, S15a and S16a**. The energy and force errors in these systems range from 1.27 meV/atom to 3.56 meV/atom and 62 meV/Å to 116 meV/Å, respectively. In the second case, *i.e.*, sets 3 and 4 in **Tables S7 and S8**, solvent molecules were equilibrated using MLP-MD before running the validation MD. The energy and force errors for water and methanol for solvated systems vary from 0.3 meV/atom to 1.8 meV/atom and 52 meV/Å to 62

meV/Å. These validations indicate that adding more solvent molecules does not deteriorate the accuracy and the potentials have sufficient transferability across systems with different sizes. The large errors for testing in entries 1 and 2 in **Tables S7** and **S8** are likely due to distorted structures in the validation trajectory, as these MDs were initiated from randomly placed solvent molecules. While testing MLPs on trajectories starting from randomly placed molecules may seem counterintuitive, this approach challenges the performance of MLPs in modelling distorted structures and out-of-equilibrium geometries.

Table R2 (S7). Errors of ACE MLPs for Diels-Alder reaction in water and methanol with testing system larger than training systems. Each set contains 101 structures generated by independent 500-fs MLP MD trajectories.

	Testing System	Water		Methanol	
		Energy MAD (meV/atom)	Force MAD (meV/Å)	Energy MAD (meV/atom)	Force MAD (meV/Å)
1	TS _{CP+MVK} + 3 waters / 2 methanol, endo	2.07	105	3.21	93
2	TS _{CP+MVK} + 3 waters / 2 methanol, exo	1.77	95	1.27	62
3	TS _{CP+MVK} + 55 waters / 40 methanol, endo	0.37	59	0.30	54
4	TS _{CP+MVK} + 55 waters / 40 methanol, exo	1.79	62	0.52	57

Table R3(S8). Errors of ACE MLPs for the Diels-Alder reaction in water and methanol for different training-like systems. Subset 2 consists of TS_{CP+MVK} + 2 water molecules/1 methanol molecule(s), while Subset 3 consists of TS_{CP+MVK} + 33 solvent molecules. The configurations used to test the MLPs were generated independently by MLP-MD and were not included in the training data.

	Training-like System	Water		Methanol	
		Energy MAD (meV/atom)	Force MAD (meV/Å)	Energy MAD (meV/atom)	Force MAD (meV/Å)
1	Subset 2, endo	2.5	88	1.61	116
2	Subset 2, exo	1.75	77	3.56	106
3	Subset 3, endo	0.48	58	0.68	52
4	Subset 3, exo	0.97	58	0.86	56

Furthermore, although tested on geometries from MP2-SCS calculations (and apparently not on energies), all methods used in the present work are far from truly quantitative reference methods such as CCSD(T). Is the "agreement with experiment" not rather fortuitous?

To identify the most suitable level of theory, we evaluated both TS geometries and energies. For TS geometries, we compared different levels of theory against the SCS-MP2 method (**Table S5 SI §S4**). Energies were calculated using different DFT functionals on IRC geometries obtained at the PBE0-D3BJ/def2-SVP level of theory, with the SCS-MP2 method serving as the reference. PBE0 was chosen

based on its reliability and computational efficiency (*J. Am. Chem. Soc.* **2020**, 142, 1300–1310). The choice of SCS-MP2 as a reference is based on its demonstrated excellent agreement with experimental data for pericyclic reactions (*Angew. Chem., Int. Ed.*, **2008**, 47, 7746-7749 and *Chem.–Eur. J.*, **2004**, 10, 6468-6475). Moreover, a previous study has also demonstrated that SCS-MP2 provide accurate reaction energies as well as TS geometries when compared to CCSD(T) for Diels-Alder reactions (*Phys. Chem. Chem. Phys.*, **2013**, 15, 5108-5114). The ω B97M-D3BJ level of theory, which exhibits the lowest deviations from SCS-MP2 in both TS geometry and energies, is therefore selected as the ground-truth method to train ACE MLPs.

2. What are the sizes of the training, test and validation sets?

We acknowledge the need for further clarification of the size and generation of datasets. This information has now been provided in the response to Reviewer #1 (**Table R1**) and **Table S6 SI §S6.1**. It is important to note that only training sets and test sets are used in this work. A validation set is often used for fine-tuning, model selection, and hyperparameter optimization, preventing overfitting of the model. In this work, we use ACE descriptors combined with linear regression, which do not require these steps. In this case, model overfitting is prevented using the ridge regression regularization technique, which includes only one hyperparameter in the model - the penalty weight - set to 0.1 (see **SI §S2**). Hyperparameter optimization was not conducted, and we relied on a value used in ref. *J. Chem. Theory Comput.* **2021**, 17, 7696–7711.

3. Are downhill trajectories really suitable for investigating the (a)synchronicity of a reaction? How much does omitting the transit from reactant to the TS affect the dynamics and therefore the (a)synchronicity? Note that the question of (a)synchronicity for DA reactions (in gas phase - i.e. comparable to simulations in "implicit solvent" here) has also been considered for 2,3-dibromo-1,3-butadiene and maleic anhydride, see Rivero et al. (*JCP* 2019; *Mol. Phys.* 2020).

The use of downhill trajectories propagated from the TS has been widely employed in exploring reaction mechanisms, especially in cases where dynamic effects are significant and traditional transition state theory (TST) fails to predict selectivity. In such instances, quasiclassical trajectories are propagated from the previously determined TS through static QM calculations. The capabilities of this approach in elucidating reaction mechanisms have been demonstrated in pioneering works by K. Houk, D. J. Tantillo and D. Singleton, to name a few. Examples include reactions with post-transition state bifurcations (PTSB), where a single TS “bifurcates” without the intervention of an intermediate to yield two products. Noteworthy investigations include cyclopentadiene dimerization extensively investigated by Houk and colleagues (K. Houk *et al.*, *Angew. Chem. Int. Ed.*, **2008**, 47, 7592; D. J. Tantillo, *Pure and Applied Chemistry*, **2017**, 89, 679; D. J. Tantillo, *J. Phys. Org. Chem.* **2021**; 34:e4202), carbocation reactivity (Tantillo *et al.*, *J. Am. Chem. Soc.* **2021**, 143, 2, 1088) and hydroboration of terminal alkenes to determine the origin of the observed experimental selectivity (Singleton *et al.*, *J. Am. Chem. Soc.* **2017**, 139, 44, 15710–15723).

Downhill dynamics have also been widely applied in reactions with various degrees of synchronicity. For example, the effects of solvent and enzyme on the product ratio and (a)synchronicity of the Diels-Alder reactions forming *spinosyn A* have also been revealed by the downhill dynamics (Houk *et al.*, *PNAS*, **2018**, 115, 5, E848-E855). R. L. Longo *et al.* explored the synchronicity of Diels-Alder reactions through both static and downhill dynamics approaches in the gas phase (Longo *et al.*, *J. Comput. Chem.* **2016**, 37, 701–711). They demonstrated the importance of adopting a dynamic perspective to effectively classify and quantify the (a)synchronicity of these reaction mechanisms, highlighting the severe limitations of the static approach. Our previous works (Duarte *et al.*, *Chem. Sci.*, **2021**, 12, 10944-10955; Duarte *et al.*, *Phys. Chem. Chem. Phys.*, **2022**, 24, 20820) have also demonstrated the usefulness of downhill dynamics in investigating product ratios for reactions with bi- or tri- furcations.

Despite these examples, we acknowledge the limitations associated with downhill dynamics, particularly their dependency on prior knowledge of the TS region and the conditions used for propagation (e.g., temperature and zero-point energy considerations).

While we agree with the reviewers that the synchronicity of Diels-Alder reactions has been extensively explored, most studies have been conducted in the gas phase or utilized implicit solvent models due to computational costs. In contrast, our work extends beyond the PES to explicitly consider the free energy surface in the solvent. This demonstrates the promise of MLPs in overcoming the limitations of QM-based dynamics, offering a robust and comprehensive approach to elucidate reaction mechanisms of increasingly complex systems. By leveraging the strengths of both uphill and downhill dynamics, we believe our methodology offers a robust and comprehensive means to elucidate reaction mechanisms, ensuring a thorough exploration of the (a)synchronicity in the context of dynamic organic processes.

4. Similar effects as to transient formation of the zwitterion were also reported for double proton transfer in formic acid in solution using machine learned PES, see Töpfer et al., PCCP 2022.

We thank the reviewer for bringing this paper to our attention. We added a reference to the work of Töpfer *et al.*, *Phys. Chem. Chem. Phys.* **2022**, 24, 13869-13882 to SI §S8 [ref. 36] to stay within the reference limit required by the journal.

5. To put the energetics discussed in the main MS and in the SI in perspective, using identical units throughout will be helpful.

Throughout our manuscript, we consistently employ two unit sets: kcal/mol for reaction energetics and meV/atom and meV/Å for energy and force errors, respectively. This choice is deliberate, considering that it is common practice within the chemistry community to discuss reaction energetics and barriers in kcal/mol. Conversely, the MLP community often reports training errors for complex condensed-phase systems in eV. We have retained this approach as it allows for meaningful comparisons of training errors across systems of varying sizes and different MLP models. We believe the distinct nature of the studied properties justifies this unit selection.

6. The validation in the SI for energies and forces appears to have been carried out for idealized scenarios ("gas-phase TS geometry immersed in 54 water molecules, which was equilibrated by 1 ps MD simulation with fixed TS before propagating"). Would it not be more natural to just pick samples from equilibrium or the US trajectories?

The choice of using a different system for validation was primarily driven by practical considerations. We opted for this approach because of the relatively low computational cost associated with MLPs. This allowed us to conduct extensive US free energy calculations in large systems, which included solutes in the presence of 200 water and 90 methanol molecules, ensuring a proper description of solvation. However, such large systems would be computationally demanding for DFT. As a result, the validation of our approach was performed in smaller model systems, comprising 55 water and 40 methanol molecules. These smaller systems were chosen to strike a balance between computational feasibility and the need for a meaningful validation of our approach.

Reviewer #3 (Remarks to the Author):

1. When the authors discuss the state-of-the-art of MLPs for solvation in the Introduction (specifically in the first paragraph in page 3), the authors miss to refer to relevant reference, for instance, <https://pubs.aip.org/aip/jcp/article/155/8/084101/1018119/Machine-learning-implicit-solvation-for-molecular>. It would be good if the authors could look again for references in case there

are other previous works that might be useful to highlight even more the breakthrough that the present work represents.

We thank the reviewer for the comment. Unfortunately, due to space constraints, we initially omitted some relevant works. We have now revisited the introduction to incorporate additional references, acknowledging their significance in the field. Furthermore, to provide a more comprehensive overview of studies utilizing MLPs for investigating chemical processes in solvents, we have included an additional section in the SI for interested readers.

The revised text now includes references to works that have employed MLPs to investigate chemical processes in solvents. Several approaches have been employed to account for the solvent effects including: i) employing implicit solvent (Noé *et al.*, *J. Chem. Phys.* **2021**, 155, 084101; Riniker *et al.*, *J. Chem. Phys.* **2023**, 158, 204101); ii) implementing of Δ -learning schemes to improve QM/MM calculations (Yang *et al.*, *J. Chem. Theory Comput.* **2016**, 12, 4945; Elstner *et al.*, *J. Chem. Theory Comput.* **2022**, 18(2):1213); iii) using of ML/MM frameworks for reactions in explicit solvents (Meuwly *et al.*, *Phys. Chem. Chem. Phys.* **2022**, 24, 13869; Xie *et al.*, *J. Chem. Theory Comput.* **2023**, 19, 1157); or iv) employing of directly trained MLPs for reactions in explicit solvents (Yang *et al.*, *Chem. Eur. J.* **2019**, 23, 8289; Parrinello *et al.*, *Catal. Today* **2022**, 387:143-149; Saitta *et al.*, *J. Chem. Theory Comput.* **2022**, 18(9):5410-5421). Isayev and Roitberg have recently reviewed the field in Roitberg *et al.* (ChemRxiv. 2023, DOI: 10.26434/chemrxiv-2023-x82fz).

We have made these additions aiming to strike a balance between providing a thorough literature background and adhering to the reference limit. It is important to emphasise that while there are works utilizing reactive MLPs with explicit solvent models, they typically involve computationally expensive AIMD with >10,000 configurations to train NNs, whereas our approach employed fewer than a thousand configurations achieving significant computational efficiency compared to existing methods. The modified text in the Introduction of the main manuscript on page 1 is as follows:

While there are prominent examples of using MLPs to correct QM/MM computed free energies [29], modelling solvent implicitly [30, 31] or replacing the QM part to have a more efficient approach, ML/MM,[32] only a handful of examples exist where chemical processes have been modelled in explicit solvent fully using MLPs. Such examples include urea decomposition in water [33], 1,3-dipolar cycloadditions in water [34], Strecker-cyanohydrin synthesis of glycine in water [35], and reactions in alkali carbonate–hydroxide electrolytes. [36] In all these cases, NNPs are used, requiring thousands of AIMD configurations. For a comprehensive overview of relevant works, we refer the reader to a recent review by Isayev and Roitberg ref [37] and **SI §S1**.

[29] Yang *et al.*, *J. Chem. Theory Comput.* **2016**, 12, 4945-4946.

[30] Noé *et al.*, *J. Chem. Phys.* **2021**, 155, 084101.

[31] Shao *et al.*, *RSC Adv.*, **2023**, 13, 4565-4577.

[32] Xie *et al.*, *J. Chem. Theory Comput.* **2023**, 19, 1157.

[33] Parrinello *et al.*, *Catal. Today* **2022**, 387,143-149.

[34] Yang *et al.*, *Chem. Eur. J.* **2019**, 23,8289-8203.

[35] Saitta *et al.*, *J. Chem. Theory Comput.* **2022**, 18, 5410-5421.

[36] Panagiotopoulos *et al.*, *J. Chem. Theory Comput.* **2023**, 19, 4584–4595.

[37] Roitberg *et al.*, ChemRxiv. DOI: 10.26434/chemrxiv-2023-x82fz.

The additional section on MLPs and solvents in SI (**§S1**) is as follows:

MLPs have been developed to simulate various chemical processes in solution, including modelling of peptides, spectroscopy analyses, and chemical reactions. As far as the solvent effects are concerned, most systems have focused on employing implicit solvent while few have considered the solvent

explicitly either using MLPs in conjunction with QM/MM methods or describing the whole system with MLPs. Isayev and Roitberg have recently reviewed the field in ref [1]. We provide here a summary of the most relevant works on the application of MLPs in modelling solvation effects.

Implicit Solvent Models using MLPs:

- Noé, Clementi and colleagues [2, 3] developed an MLP-based implicit solvent model for peptides, trained on configurations from explicit-solvent MD simulations. Shao *et al.* [4] applied a similar approach, using the DeepPot-SE representation to define the features of the solute structure. In addition to MM configurations, their training also included *ab initio* QM modelling of the solvated molecule.
- Müller and coworkers [5] introduced FieldSchNet, an MLP method capable of describing the interactions between molecules and an external field. By considering the solvent as a continuum external field with a specific dielectric constant, FieldSchNet is suitable for modelling molecules in a continuum solvent. Furthermore, the model is applicable within the ML/MM approach, as discussed below.
- Riniker *et al.* [6] introduced a graph neural network (GNN)-based implicit solvent model to simulate the dynamics of peptides. The main purpose of the model is to decrease the number of degrees of freedom compared to dynamics in explicit solvent and accelerate the sampling. The GNN model improves the base GB-Neck2 model within an Δ -learning scheme by calculating the solvation forces acting on the peptides.

Explicit Solvent with Δ -learning schemes:

- Yang *et al.* [7] utilized a Behler and Parrinello-type HDNNP to predict QM/MM potential energies for an SN₂, proton transfer of glycine, and the Claisen rearrangement reactions in explicit solvent. They first performed semiempirical QM/MM simulations, and then the free-energy profile along free-energy obtained at the SQM/MM was reweighted with NN predicted potential energies to enhance accuracy. The QM atomic charges at the SQM/MM level are introduced to NN to capture the polarization of the QM subsystem induced by the MM environment.
- Riniker and colleagues [8] employed a Δ -learning scheme to uplift the QM energies from DFTB to various DFT methods in QM/MM simulations. The resulting potentials were validated by performing ML/MM MD simulations of retinoic acid in water and the interaction between S-adenosylmethioniate and cytosine in water.
- Corminboeuf and Ceriotti [9] trained direct and Δ -learning Behler and Parrinello-type NNPs to reproduce energies and forces at the PBE0-D3BJ level of theory either directly, or by correcting the DFTB approach. Both models were combined within a multiple-time step algorithm to stabilize direct NNP and decrease the cost of the delta model. The resulting approach was applied to simulate the challenging properties of methanesulfonic acid in a complex mixture of phenol and H₂O₂ and investigated the role of nuclear quantum effects in hydrogen bonding. The same scheme was used to train direct and delta NNPs (GFNO-xtb to PBE0-D3BJ) to identify prevalent non-covalent interactions in a benzotelluradiazole-Cl complex in an explicit THF solvent.[10]

Direct MLP for Explicit Solvent with ML/MM Scheme:

- The FieldSchNet model introduced by Müller and coworkers [5] was used within an ML/MM approach to simulate molecular spectra accounting for solvent effects. Here, the QM region is entirely replaced by MLPs while the solvent is considered explicitly using electronic embedding to couple the ML and MM regions.
- Meuwly and the Xie group [11] employed PhysNet ML/MM to investigate the double proton transfer within the hydrated formic acid dimer in explicit water; the solute was modelled at the MP2 level and water at the MM level. Water solvent was found to promote the first proton transfer through a favourable solvent-induced Coulomb force along the O–H \cdots O hydrogen bond,

while the second proton transfer was significantly controlled by the O–O separation and other conformational degrees of freedom.

- Xie group [12] adopted a molecular embedding ML/MM scheme to simulate an SN2 reaction in water, further correcting the results using a weighted thermodynamic perturbation (wTP) scheme.

Direct MLP for the Whole System Including Explicit Solvents

- Saitta *et al.* [13] proposed an ab initio protocol using DeepMD NNPs, studying the Strecker-cyanohydrin mechanism for glycine synthesis in water solution.
- We have used a kernel-based Gaussian approximation potential (GAP) to study an SN2 reaction in solvent using a small dataset, achieving high accuracy, and analysing the importance of solvent effects by comparing the difference between trajectories in implicit and explicit solvents. [14]
- Parrinello *et al.* [15] applied AIMD combined with metadynamics and several extra active learning loops to train a DeepMD for urea decomposition in explicit water.
- Yang and coworkers [16] trained a TensorMol NNP for two 1,3-dipolar cycloaddition reactions in explicit water solvent and ran MLP-MD downhill dynamics to investigate the mechanism.

Applying MLPs to chemical reaction modelling in a condensed phase is fast-moving field of great significance for the modelling community, as the accurate description of solvent effects remains one of the grand challenges in computational chemistry. The papers mentioned above, particularly those utilizing NN-based potentials, typically require thousands of data points to obtain reliable models. In contrast, our approach, employing descriptor-based selectors combined with linear ACE, can train MLPs for the entire solvated system using only several hundred data points. This technique has the potential to advance the developments in this field, pushing the boundaries of modelling processes in solution.

[1] Roitberg *et al.*, ChemRxiv. DOI: 10.26434/chemrxiv-2023-x82fz.

[2] Noé *et al.*, *J. Chem. Phys.* **2021**, 155, 084101.

[3] Clementi *et al.*, *Annu. Rev. Phys. Chem.*, **2020**, 71, 361–390.

[4] Shao *et al.*, *RSC Adv.*, 2023, 13, 4565-4577.

[5] Müller *et al.*, *Chem. Sci.*, **2021**, 12, 11473-11483.

[6] Riniker *et al.*, *J. Chem. Phys.* **2023**, 158, 204101.

[7] Yang *et al.*, *J. Chem. Theory Comput.* **2016**, 12, 4934–4946.

[8] Riniker *et al.*, *J. Chem. Theory Comput.* **2021**, 17, 2641–2658.

[9] Corminboeuf *et al.*, *J. Chem. Theory Comput.* **2020**, 16, 5139–5149.

[10] Corminboeuf *et al.*, *J. Chem. Phys.* **2022**, 156, 154112.

[11] Meuwly *et al.*, *Phys. Chem. Chem. Phys.* **2022**, 24, 13869-13882.

[12] Xie *et al.*, *J. Chem. Theory Comput.* **2023**, 19, 4, 1157–1169.

[13] Saitta *et al.*, *J. Chem. Theory Comput.* **2022**, 18, 9, 5410–5421.

[14] Duarte *et al.* *Chem. Sci.* **2021**, 12, 10944–10955.

[15] Parrinello *et al.*, *Catal. Today* **2022**, 387, 143-149.

[16] Yang *et al.*, *Chem. Eur. J.* **2019**, 23, 8289-8203.

2. At the end of page 4 the authors say: In contrast to the metrics described above, these selectors use molecular descriptors, such as Smooth Overlap of Atomic Positions (SOAP) or ACE. To assess whether the training set accurately represent the chemical space of interest. This sentence makes the reader to think that the authors will use both descriptors to construct their selector, but later in the text it seems that only SOAP was used for such task. Could the authors please clarify this point and maybe rewrite part of the sentence to avoid confusions?

The reviewer is correct, while either SOAP or other descriptors could be used in combination with the *similarity* and *distance* selectors, we only used the SOAP descriptor. This decision was motivated by the widespread application and successful implementation of the SOAP descriptor in numerous systems, making it an appropriate choice for testing our selection scheme. To avoid the confusion, we modify the paragraph on page 2 of the main text as follows:

In contrast to the metrics described above, these selectors use molecular descriptors, such as Smooth Overlap of Atomic Positions (SOAP), to assess whether the training set accurately represents the chemical space of interest. By examining the SOAP descriptor space for the structures in the training set, we demonstrate that these selectors provide a general metric applicable across different MLP approaches at a low computational cost.

3. The authors proved in their work that the descriptor-based selectors outperform the typical energy-based ones. However, I would like the authors to spend a few sentences to discuss the possible drawbacks that can become relevant once their approach is used in larger (and possibly more flexible) systems. Currently, few examples come to my mind:

i) Descriptors such as SOAP, and even ACE, neglect all (or at least some) long-range features. For larger systems, it could be the case that molecular geometries share very similar local environments, while their long-range features are different. In such a case, the descriptor-based selector could indicate that these configurations are similar when they are not.

ii) Even for medium-sized systems, the authors of the original SOAP paper themselves have proved that the SOAP descriptor can have problems with its "uniqueness" (i.e., two geometries that are different share the exact same SOAP descriptor). This could be a great asset for the Conclusions/perspectives of the work.

We appreciate the comment regarding the potential limitations associated with the descriptor-based selectors employed in this study. In line with existing literature, the limitations highlighted by the reviewer can be attributed to two factors. The first limitation arises from the geometrical incompleteness of the SOAP descriptor, which is based on 3-body features. The structural degeneracy can be addressed by considering SOAP descriptors including all elements in the system instead of solely considering the descriptor of one element (Ceriotti *et al.*, *Phys. Rev. Lett.* **2020**, 125, 166001; Ceriotti *et al.*, *Chem. Rev.* **2021**, 121, 9759-9815). In our study, we computed SOAP descriptors for all elements present in the systems, thus mitigating the issue of incompleteness. However, for highly complex systems, it may be advantageous to consider alternative descriptors that incorporate high-order features, such as ACE descriptors. These descriptors offer better geometrical completeness and are natural replacements for the SOAP with no additional efforts, as the underlying principles of selectors remain valid. The second limitation stems from the locality assumption of the SOAP descriptor, meaning it may fail to distinguish between chemical environments that only differ in a long-range region beyond the cut-off radius. This limitation can be overcome by combining the short-range representation, such as SOAP, with long-range descriptors, such as the long-distance equivariant (LODE) framework proposed by Ceriotti and colleagues (*J. Chem. Phys.* **2019**, 151, 204105, *J. Phys. Chem. Lett.* **2023**, 14, 9612–9618). Despite a number of developments in the field, the inclusion of long-range interactions in the MLPs remains an actively discussed problem. Isayev and colleagues have recently reviewed the use of MLPs to capture non-local interactions, where several models utilize self-consistency or message-passing layers (Isayev *et al.*, *J. Phys. Chem. A* **2023**, 127, 11, 2417-2431).

Both injective and concise descriptors, as well as long-range MLPs, are promising avenues currently under development in our group. In future work, we hope to update the descriptor-based selectors

to account for such factors. To reflect on this, we have added the following paragraph to the discussion section of the main text, page 10:

This work applies the SOAP approach, which is based on 3-body features, for the descriptor-based selectors. Although we demonstrated the advantages of using SOAP-based selectors in automated AL, these selectors may face challenges when applied to highly complex systems due to their geometrical incompleteness. In such cases, descriptors with higher-order features, such as ACE, would serve as a suitable replacement for the SOAP descriptors within the same framework. Another potential limitation arises from the locality assumption used in the SOAP. This means that the descriptor-based selectors may be unable to differentiate between chemical environments that only differ in a long-range region beyond the descriptor cut-off. In such cases, employing a larger cut-off or combination of short-range and long-range descriptors would be beneficial. The use of both injective and concise descriptors, as well as the development of long-range MLPs, are areas currently being explored. In future work, we aim to update the descriptor-based selectors in the *mlp-train* code following these advancements.

4. Figure 1 misses the letter "a" in Top Figure.

We corrected the figure accordingly.

5. In page 7, in the last paragraph of the "Workflow" subsection, the authors mention that the time of the simulation is set to $(n^3 + 2)$ fs, but the timestep is missing. I consider that it is relevant in order to know how many configurations are sampled in each run.

The time step for the MD is 0.5 fs, as specified in the Computational details, ACE training section. To clarify this, we modified the sentence in the main text:

The simulation time is set to $(n^3 + 2)$ fs where n corresponds to the index of the MD run, starting from 0. The time step of the dynamics is 0.5 fs.

6. In page 8, the authors mention that: Selecting an appropriate threshold is key as too low values (e.g., similarity below 0.9) can result in the selection of non-physical structures that fail to converge in the self-consistent field (SCF) computations, while too high values (e.g., 1) do not provide any additional information. With respect this sentence:

i) Does this mean that you only select configurations with similarity values between 0.9 and 1?

Indeed, only structures falling within an established similarity range are incorporated into the training data set. For the system under study, our threshold was set at 0.999, therefore only structures with similarity values between 0.999 and 1.000 were selected. This threshold can be adjusted as the potential stabilizes throughout training.

ii) Do you obtain non-physical structures from the MDs run with your MLP? If so, what do you do within your framework to deal with this problem: do you select the last "physical" structure available?

Some non-physical structures are indeed generated during active learning, especially at the early stages of the training when MLPs do not have sufficient information on the repulsive and dissociative regions. While we have found that retaining a small number of relatively high energy structures is necessary when training reactive MLPs, to better describe the PES regions away from the minima, the presence of highly distorted structures away from the chemical space of the process being explored can deteriorate the quality of the potential. In this case, for example, we have retained some high-energy structures where solvent molecules in proximity are distorted; however, we eliminated some

early structures where SCF calculation failed. To address these challenges, we have implemented a backpropagating algorithm, which retraces the trajectory, identifying the first structure where the selection criterion surpasses the threshold set by the selector.

7. The authors use the Euclidean distance as metric in the calculation of the local outlier factor (LOF). It is known that in high-dimensional spaces (such as the one spawned by SOAP vectors) the Euclidean metric is not as reliable as in lower-dimensional spaces. Have the authors explored the use of other metrics in the calculation of the LOF? Maybe for the example proposed in the present work there would not be differences, but for other (larger) systems it could be relevant. Do the authors consider that this could be an issue for extending the application of their approach to other systems of interest?

We acknowledge the potential shortcomings of Euclidean distance due to the curse of dimensionality. Consequently, we have updated our *distance* selector by performing dimensional reduction of SOAP space before selecting outliers, which is discussed in the following paragraphs.

In our initial evaluation of *distance* selectors when training the water system, we also evaluated the performance of the cosine similarity (A. Singhal, *Bulletin of the IEEE Computer Society Technical Committee on Data Engineering* **2001**, 24, 35–43.) and Manhattan distance (E. F. Krause, *Taxicab Geometry*, **1987**, Dover, ISBN: 978-0-486-25202-5.), alongside the Euclidean distance (**Table R4**, now included in **SI §S3.2**, **Table S3**). No significant differences in data efficiency and accuracy of the resulting MLPs were observed. We attribute the similar performance to the relative simplicity of the water system and to the fact that we employed the proportion of the outliers in the dataset as a selection criterion rather a fixed threshold. This approach is explained in more detail below in reply to comment 8.

Table R4. Comparisons of the performance of distance selector with different distance metrics for systems with 27 water molecules

Distance Metrics	# Configs.	Errors	
		Energy MAD (meV/atom)	Force MAD (meV/Å)
Euclidean	52	2.69	89
Cosine	56	3.49	80
Manhattan	73	3.40	75

We opted for the Euclidian metrics in the *distance* selector for its simplicity and intuitive interpretation in Euclidian space. Recognizing the importance of versatility when dealing with sparse or high-dimensional data, we have expanded the choices available distance selector in our `mlp-train` code to include the option of using either cosine similarity or Manhattan distance (see **SI §S3.2**)

We have added the following paragraph to the SI:

In the analysis presented above, the distance selector uses the Euclidean distance as a metric. We also evaluated the performance of the cosine similarity (A. Singhal, *Bulletin of the IEEE Computer Society Technical Committee on Data Engineering* **2001**, 24, 35–43.) and Manhattan distance (E. F. Krause, *Taxicab Geometry*, **1987**, Dover, ISBN: 978-0-486-25202-5.) as alternatives to the Euclidean distance (Table R4). No significant differences in data efficiency and accuracy of the resulting MLPs were observed. We attribute the similar performance to the relative simplicity of the water system and to the fact that we employed the proportion of the outliers in the dataset as a selection criterion, as

opposed to a simple fixed threshold. Therefore, we chose the Euclidian metrics for the *distance* selector due to its simplicity and intuitive understanding of the distance in Euclidian space.

To further address concerns related to high-dimensional datasets, we have incorporated dimensionality reduction using principal component analysis (PCA) before calculating the LOF in the distance selector (see M. E. Houle *et al.*, in *2010 IEEE International Conference on Data Mining*, Sydney, NSW, Australia, **2010**, pp. 128-137 and Faisal, S. *Information Technology & Control*. **2021**, 50, 138-152.). This process involves reducing the SOAP descriptor space to its three most significant directional dimensions. Subsequently, outlier identification using LOF is conducted in this 3-D space with Euclidean distance metrics. We applied this updated approach to the gas-phase reaction of CP and MVK, comparing its performance with the original distance selector.

The results, illustrated in **Fig. R4** and **Table R5** (included in the SI as **Fig. S5** and **Table S4**), demonstrate that the updated selector exhibits more efficient LOF computation, requiring fewer training data points (123 points vs. 252 points), while maintaining comparable accuracy. Although the updated selector shows slightly higher errors (0.54 meV/atom vs. 1.18 meV/atom **Fig. R4**), we attribute this to recrossing events in the validation trajectory, i.e., return to the reactants as seen from the increase in energy, used to test the updated *distance* selector, rather than inherent deficiencies in the updated approach.

To obtain the performance of the updated selector on a comparable data set, we evaluated the updated selector using the configurations generated by the original selector. Both distance selectors exhibit comparable accuracy, as indicated in **Table R5**, with energy errors of 0.55 and 0.36 meV/atom for the original and updated distance selectors, respectively. We have now included **Fig. R4(S5)**, **Table R5 (S4)** and the following paragraph in the SI (**SI §S3.2**):

Distance metrics and, consequently, *distance* selectors, may be affected by the curse of dimensionality. To address this issue, the *distance* selector includes an option to apply principal component analysis (PCA) for reducing the descriptor space before outlier detection. The MLPs in the gas phase reaction of CP and MVK are shown in **Fig. S5** and **Table S4**, comparing the MLPs trained with the original *distance* selector and the updated one with PCA. Both *distance* selectors produce accurate MLPs. However, in cases where the system is highly complex, the updated *distance* selector is more suitable and is therefore recommended.

Figure R4 (S5). Performance of distance selector for the endo reaction of CP and MVK in the gas phase. Comparisons of energies and forces between ground-truth (ω B97M-D3BJ/def2-TZVP) and ACE employing (left) original (without PCA) and (right) updated (with PCA) distance selector over 200-fs ACE MLP-MD trajectories started from TS (300 K, time step = 0.5 fs).

Table R5 (S4). Energy and force errors for the two versions of distance selectors evaluated on the configuration set containing 27 water molecules.

	Energy MAD (meV/atom)	Force MAD (meV/Å)
Original distance selector	0.55	50
Updated distance selector	0.36	30

8. In the previous-to-last paragraph of page 9, the authors mention: "we chose the threshold to encompass 20% of the outliers present in the training data". This is not clear to me. What does this mean? How do they define/select the outliers in the training?

The Local Outlier Factor (LOF) (Eqn. 4 in the main text) is determined based on the distance density of a point from its neighbours. A LOF value ≤ 1 signifies an inlier point located inside a cluster, while a LOF value > 1 indicates an outlier, i.e., a point lying in a region with a lower density of the points. Establishing a universally valid LOF threshold to define an outlier poses a challenge, given the diversity of datasets.

To address this variability, we adopted a dynamic approach. Specifically, we have chosen to set our threshold as the 80th percentile of the LOF values obtained from the training data. This means the threshold is a LOF value larger than 80% of the LOF values computed for each point of the training data set. This approach ensures that the threshold value adapts to the PES exploration during the AL iterations. If the LOF value for a tested structure is greater than or equal to the LOF threshold, it is recognized as an outlier. Alternative approaches exist to address this challenge, such as those incorporating ensemble learning or statistical scaling processes (Sander *et al.*, *ACM SIGKDD Explor. News.* **2014**, 15, pp. 11-22; Zimel *et al.*, In *Proc. ICDM 2011*, 13-24). However, their effective application typically requires much larger data sets than presented here, i.e., several hundred to thousands of data points, and their performance on small datasets has not been evaluated (Sander *et*

al., *ACM SIGKDD Explor. Newsl.* **2014**, 15, pp. 11-22). To clarify this, we modified the sentence in the main text:

Since there is no definitive rule for selecting an LOF threshold to identify outliers, in this study, we chose the threshold as an LOF value larger than 80% of LOF values for the given training set. This approach ensures that the threshold value varies with the PES exploration during the AL iterations.

9. In terms of data availability, it would be good if the authors also make their datasets available.

We have made our datasets available through the Oxford Research Archive (ORA) and included this information in the **Data Availability** section. Since the university provides a DOI and open access link to the data only upon publication, we have temporarily provided a Zip folder with all the files, including a readme file. Additionally, we are actively working on creating a comprehensive library of solvent structures, which will be available via `mlp-train` to simplify the training of systems in explicit solvent.

Reviewer #4 (Remarks to the Author):

1. The authors show enough evidence to prove that their methods are more efficient compared to the energy baseline using bulk water as a simple study case. The authors also claim in the Conclusion that the MLPs obtained with their selectors are more accurate than those obtained with energy. I agree with the efficiency claim because the number of conformations required by similarity and distance was much lower compared to the baseline, but the energy MAD error for similarity was higher than that of energy (Figures S1-S2). Since similarity was used throughout the study, the authors should note in the Conclusion that the energy MAD for energy was lower than that for similarity and then explain if the difference is negligible or not.

Maybe another way to run this comparison, if possible, would be to restrict the total number of conformations that energy can train on to match the final number of conformations selected by similarity (40) and then compare the MADs. I would expect the MADs for energy to be higher because the conformation set would be less diverse. I think this comparison would showcase the advantage of similarity, which would hypothetically be more accurate than energy with an equal number of training samples (but not in a more general sense).

We thank the reviewer for their suggestions. In response, we have conducted a detailed comparative analysis between our similarity and energy selector.

As rightly pointed out by the reviewer, the *energy* selector exhibits a lower MAD in total energy prediction than the *similarity* selector in the original manuscript, 2.89 meV/atom vs. 6.31 meV/atom, when using random initial configuration (**Table R6 (S2)** and **SI §S3.1**). However, the force MAD is lower for the similarity selector (92 meV/Å vs. 59 meV/Å, **Table R6**), which also has a higher R^2 ($R^2_{\text{Energy}}=0.990$ vs. $R^2_{\text{Similarity}}=0.993$ for energy and $R^2_{\text{Energy}}=0.970$ vs. $R^2_{\text{Similarity}}=0.987$ for forces using *energy* and *similarity* selectors, respectively, **Fig. S1** and **S2**). The close to 1 R^2 values for similarity, together with the demonstrated accuracy of force predictions, which provide vital information about the local curvature of the PES, suggest that the higher MAD for the *similarity* selector corresponds to a shift in absolute energy, which does not impact the stability of the potential nor the computed relative energies.

It is pertinent to note that the validation of the MLPs was performed on MLP-MD trajectories of 27 water molecules initiated from randomly placed configurations in a periodic box, which may include distorted structures. These structures significantly differ from the training set and can lead to higher MAD in energy and forces than for structures closer to equilibrium. However, such distorted structures

are not typical in regular MD runs because they are initiated from equilibrated structures, and the corresponding errors can be considered an estimate of the maximum expected errors. To address concerns about the influence of using too-distorted structures as a starting point on the reported MADs, we performed additional validation on the 27 water molecules system, using configurations generated by classical MD using the TIP3P water model. The errors of MLP trained by each selector are listed in **Table R6**, highlighting the significant improvement in MAD compared to randomly placed structures, particularly for the *similarity* selector.

We have followed the reviewer’s suggestion to restrict the total number of conformations for the energy selector to match the final number selected by the *similarity* and *distance* selectors (40 structures). The resulting MLP was not stable enough to propagate MLP-MD, leading to instabilities in the MD after 100 fs, highlighting the better performance of the *descriptor-based* selectors in providing a stable and accurate representation of the potential energy landscape.

In conclusion, this analysis supports the assertion that while the energy selector may exhibit a lower MAD in total energy prediction, when compared using the same number of configurations generated in a similar manner, the similarity selector provides a much more stable potential at lower cost.

We have now included the following paragraph in the SI (**SI §S3.1**):

It is worth noting that the reported validation of the MLPs was performed on MLP-MD trajectories of 27 water molecules initiated from randomly placed configurations in a periodic box, which may include distorted structures significantly differing from the training set. However, such distorted structures are not typical in regular MD runs because they are initiated from equilibrated structures, and the corresponding errors can be considered an estimate of the maximum expected errors. To address concerns about the influence of using too-distorted structures as a starting point, on the reported MADs, we performed additional validation on the 27 water molecules system, using configurations generated by classical MD using the TIP3P water model. The errors of MLP trained by each selector are listed in **Table S2**, highlighting the significant improvement in MAD compared to randomly placed structures, particularly for the *similarity* selector.

Table R6 (S2). Errors of ACE MLPs trained with different selectors for bulk water. The initial configuration used to run the corresponding MLP-MD to generate test data included random water in a box and equilibrated water configurations.

	Random initial configuration		Equilibrated initial configuration	
	Energy MAD (meV/atom)	Force MAD (meV/Å)	Energy MAD (meV/atom)	Force MAD (meV/Å)
energy	2.89	92	2.14	85
similarity	6.31	59	1.86	81
distance	2.69	89	2.31	76

2. About the decision not to continue using distance, did you run any tests with the DA system, either in implicit or explicit solvent, before discarding this option? There could be a more thorough explanation of how much more initial data distance requires in comparison to similarity. The MLP used to study the reaction is the one obtained at the end of the entire training process, so I don’t understand why the lack of initial data would be an impediment. Maybe distance performs worse during the early iterations of the AL cycle due to the initial data requirement but does this tell us anything about the performance of the final MLP?

Our decision not to continue using the *distance* selector does not diminish its efficacy. Indeed, the final MLP generated with this selector proves to be accurate and robust. We recommend employing this selector, particularly the updated version, when a sufficient number of initial configurations are available. In the updated version, the original distance selector is combined with Principal Component Analysis (PCA), as detailed in **SI §S3.2** and our response to reviewer 3, comment 7.

Before opting for the *similarity* selector, we conducted tests with the original *distance* selector on the *endo* reaction of CP and MVK in the gas phase. This resulted in 252 data points and an MLP with MAD of 0.5 meV/atom and 50 meV/Å for energies and forces, respectively. An important consideration in utilising the *distance* selector is the computation of the LOF, a metric of the local density that requires information on the neighbouring points. We found that, for the *distance* selector to perform optimally a minimum of 20 initial configurations is necessary for the first AL iteration. This is due to the fact that the performance of the *distance* selector improves with an increasing number of structures. With only 10 initial configurations, the *distance* selectors tend to collect unphysical structures in the first two AL iterations due to insufficient information. In contrast, the *similarity* selector proves effective with just 10 initial structures. While both selectors eventually yield accurate final MLPs with comparable MAD, our preference for the *similarity* selector in this study is rooted in practical considerations. Specifically, the similarity selector requires fewer initial configurations translating into a lower computational cost. This is especially advantageous in explicit solvent simulations.

3. The ACE MLP used to study the DA reaction in explicit water was obtained by training it with a total of 600 conformations belonging to four different subsets. How were these subsets obtained? My understanding is that the conformations were chosen using the similarity selector, but maybe I am misinterpreting how you set up the experiment. This could be clarified in the text or the SI.

The individual subsets were generated through an active learning scheme using the *similarity* selector. The overall size of the training data set was then adjusted to yield exactly 600 structures because of hardware restraints. This information has now been provided in the response to Reviewer #1 (**Table R1** of this report) and the **SI §S6.1, Table S6**.

4. The authors claim that their selection strategies save time in comparison to energy. I am convinced that this is the case. However, as the authors explain, there are two reasons why their strategies save time: fewer conformations need to be collected before the MLP converges to <1 kcal/mol MAD, and QM calculations are avoided. What is the relative contribution of each factor to the time saved?

Using as a case study a box containing 27 water molecules, we present below a detailed breakdown of computational demands and time efficiency when using either the *energy* or *similarity* selector.

For the *energy* selector, which requires overall 548 CPU hours for training (*vide infra*), the primary contributor to the cost are QM computations. Here, each single point gradient calculation at the PBE0-D3BJ/def2SVP level of theory requires approximately 0.5 CPU hours. With over 1000 computations required—281 for generating the training data and the rest for the *energy* selector evaluations—the cumulative computational time amounts to around 521 CPU hours. Additionally, the cost of sampling and training during the AL cycle, *i.e.*, 30 cycles for this selector, adds an additional 27 CPU hours. Thus, the total computational time for the energy selector is approximately 548 CPU hours. Notably, 69% of the cost corresponds to MLP validation during the active learning cycle and 25% is allocated to labelling the training data.

In contrast, the *similarity* selector requires QM computations only for the data present in the training set; which corresponds to 40 configurations for the water system. The corresponding computational time is thus approximately 20 CPU hours plus 4 CPU hours considering the 5 cycles needed during the AL loop. Thus, the *similarity* selector requires 24 CPU hours, with approximately 80% of the time dedicated to QM computations for labelling the training data.

In summary, the time savings provided by the *similarity* selector predominantly stem from the elimination of QM calculations during the selection within the AL loop, resulting in a saving of 381 CPU hours.

5. Since the AL loop ends when no more conformations are selected, it is possible for the training to stop before the conformational space has been sampled if the user chooses bad hyperparameters. Are there any suggestions for hyperparameter selection? Are the values that you chose for this study good defaults in general? If the users have to do exhaustive hyperparameter tuning, this might make the usage of your selection strategies more difficult.

While the measure of the similarity in the SOAP space is a general quantity and the optimal value is transferable across different systems, the sampling length is system-dependent. To ensure the stability of the dynamics during exhaustive sampling of the relevant conformational space, and consequently the stability of the potential, the length of sampling should cover the time scale of the process of interest. This indeed requires a nuanced understanding of the system and process of interest. Adjusting the sampling length can be easily done in our `mlp-train` code (<https://github.com/duartegroup/mlp-train>) in which users can extend training for a longer time by increasing the number of the parameter `max_md_time` in the AL scheme.

Regarding hyperparameter selection, we have provided a comprehensive list of hyperparameters and their corresponding values in the **SI §S2, Table S1**. These include the parameters for the ACE model, the linear regression model, the SOAP descriptors, and the selectors used in active learning. While the hyperparameters selected for the ACE descriptor and the linear model influence the quality of the potentials, the proper sampling of the conformational space is mainly determined by the length of the MD simulations propagated during AL and the threshold of the selector criteria. Here, we adopted a sampling time of up to 5 ps and the SOAP *similarity* threshold of 0.999, which are the default `mlp-train`. We kept these values as well as based on our previous studies of gas-phase Diels-Alder reactions, which have been found to be suitable for organic reactions in general. Nonetheless, users are encouraged to explore hyperparameter space, particularly when investigating complex processes to enhance the robustness of the active learning strategy. Moreover, we are currently working on the inclusion of enhanced sampling techniques during AL to ensure the conformational space is sufficiently sampled without the need to propagate significantly long MD simulations.

6. There seems to be no discussion about the relative burden of training the MLP vs. actually sampling the reaction pathway of a reactive system in explicit solvent. The authors claim that their workflow can be used generally to study reactions in explicit solvents, but I am unsure if the “workflow” refers simply to the AL algorithms or to the entire process of studying the reaction in explicit solvent. If the latter is the case, then I think that there are too many system-specific choices (e.g., umbrella sampling reaction coordinate) for the workflow to be widely applicable. If the former is the case, then I suggest that the authors discuss what is the advantage of using the selector-based training in the broader context of characterizing an arbitrary reaction in explicit solvent. It is possible that the actual bottleneck for a future study that applies your workflow is the sampling of the reaction pathway rather than the convergence of the MLP, and your AL algorithms only accelerate the training part.

The term ‘workflow’ in our manuscript specifically refers to the procedure of generating datasets and training the MLP. The reviewer is correct about the complexity associated with sampling reaction pathways in explicit solvent, including consideration of accuracy, computational cost, and automation.

As highlighted in our response to comment 4, the computational investigation of chemical reactions in explicitly modelled solvents using DFT is computationally demanding, and a substantial portion of the computational cost of generating MLPs arises from these calculations. To demonstrate the acceleration provided by the access to reliable MLPs in comparison to DFT-based sampling, we offer the following estimation. For the system under investigation, containing MVK and CP in 55 water molecules at the ω B97M-D3BJ/def2-TZVP level of theory, the computational demands are significant—equivalent to 2 CPUh per single point gradient computation. Our MD/US protocol, involving 30 windows with each window sampling 10 ps with a step size of 0.5 fs, spans a total of 300 ps of dynamics, translating to 600,000 gradient computations. The overall computational cost for simulating the solvated MKV + CP system using DFT-based methods would be 1,200,000 CPUh (*i.e.*, approx. 137 CPU years). In stark contrast, employing MLPs for the same computations would take around 200 CPUh. Even for the MD simulation of the larger system presented in this work, consisting of the substrate and 200 water molecules, the computational cost is only 600 CPUh. If we further account for the additional cost of MLP training, the computational gain is substantial, at least 6000-fold.

Currently, the prohibitive cost of *ab initio* MD simulations means that achieving sufficient sampling to generate converged free energy profiles and exhaustively sampling potential reaction pathways is challenging. From the presented comparison, it is thus clear that modelling fully solvated Diels-Alder reactions efficiently and accurately is achievable through the use of MLPs.

7. A minor observation: K is called the “similarity matrix” of a point, but it seems the dimensions are 1xN (where N=number of conformations already in the training set). If this is correct, it might be clearer to say K is a vector of length N.

We thank the reviewer’s comment. The changes in the main text (pg. 4) are as follows, here highlighted in bold:

During the AL loop, we apply so-called similarity- and distance-based selectors, referred here to as *similarity* and *distance* selectors, respectively. The former quantifies the similarity between a new data point p and existing configurations p' using the kernel function $k(p, p')$. **The similarity vector** of the data point is defined as:

$$K = \left(|k(p_0, p_i)|^\xi, |k(p_0, p_j)|^\xi, \dots \right)^T$$

Where p_0 is the SOAP vector of the new structure, p_i is the SOAP vector of the i -th configuration in the existing set, and ξ is a positive integer that increases the sensitivity of kernel to changes in atomic position. The kernel is computed between the new configuration and all other configurations in the training data set. The selector adds structure to the training set if the maximum value of **its similarity vector**, K , is smaller than threshold k_T , *i.e.*, $\max(K) < k_T$. Selecting an appropriate threshold is key as too low values (e.g., similarity below 0.9) can result in the selection of non-physical structures that fail to converge in the self-consistent field (SCF) computations, while too high values (e.g., 1) do not provide any additional information.

REVIEWER COMMENTS

Reviewer #1 (Remarks to the Author):

The authors have addressed all my concerns in a convincing way and exhaustively replied to the other reviewers comments. I recommend acceptance.

Reviewer #2 (Remarks to the Author):

Re-review of Zhang et al.

The authors have addressed some of the points raised. However, several issues require further attention. First notes on their replies to reviewers' comments.

R2-1: In their reply the authors mention 101 structures from independent 500 fs MLP MD trajectories. 500 fs is clearly too short to sufficiently sample conformational degrees of freedom. Either these are 500 ps or this analysis needs to be redone.

R2-1: In their reply letter they write "In the table below" - does this refer to Table R2 in the letter? (S7) probably refers to a newly added table in the SI? It would be useful to state this explicitly in the reply letter. The performance of the trained PES on true "test structures" (Table S7 entries 1 and 2) clarify that the quality of the trained NN is not particularly good. Therefore, the question still remains what the authors mean when writing "..the generated machine learning potentials exhibit excellent agreement with experimental data.." in the abstract and whether this agreement is not fortuitous given that their MLP does not generalize that well to out-of-sample structures.

R2-1: It appears that the MLP presented here is not really superior to DeepMD NN. Why would one then expect better agreement for experimental observables?

R2-2: Given that the quality of the present MLP is not superior to an DeepMD NN hyperparameter optimization would be meaningful, not?

R2-3: Downhill trajectories assume that the reaction passes through the transition state. This is not necessarily true, see e.g. recent work by Käser et al on malonaldehyde. The problem accentuates in going from the gas phase to solution because depending on the solvent structure the TS may change. In addition, downhill trajectories assume that the system crosses the TS with near-zero velocity which is also not necessarily the case. It is not good enough to state that such procedures have been "widely employed". The authors should run a forward simulation in gas phase and in solution to see how the system behaves.

R2-6: the authors should validate their MLP on the systems sizes for which DFT calculations are possible. If I understand correctly, this is MVK+CP in 55 water and 40 methanol molecules (as per the reply to R2-6).

A few additional questions:

1. The authors should demonstrate that their trajectories conserve energy.

2. It is unclear where the experimental activation free energies of 19.2 and 21.1 kcal/mol come from. Ref. 57 deals with the reaction in nonaqueous polar solvents.

3. What value for the dielectric constant was used for the simulations in implicit water?

4. Is DFT really good enough to distinguish zwitterionic from biradical character of the reaction in solvent?

In summary, there are still points to be addressed quantitatively in this manuscript before it can be recommended for publication. At this point this reviewer is not convinced that the statement "...the generated machine learning potentials exhibit excellent agreement with experimental data.." is supported.

Reviewer #3 (Remarks to the Author):

I would like to thank the authors for the thorough response to my comments (as well as to those of the other reviewers).

I think they have addressed my concerns in their entirety and updated the text accordingly. Hence, I am happy to recommend the work for publication in Nature Communications.

Reviewer #4 (Remarks to the Author):

Authors have fully addressed all the comments from the first round of review. I congratulate them for performing the excellent research presented in this manuscript.

Reviewer #4 (Remarks on code availability):

I tested the python notebooks and they worked correctly. The data and code for generating figures also looked fine. Authors could provide comments at the top of the .py files to explain the goal and the inputs/outputs of the code.

Reviewer #5 (Remarks to the Author):

Response to Reviewer Comments

Reviewer #1

The authors have addressed all my concerns in a convincing way and exhaustively replied to the other reviewers comments. I recommend acceptance.

Reviewer #3

I would like to thank the authors for the thorough response to my comments (as well as to those of the other reviewers).

I think they have addressed my concerns in their entirety and updated the text accordingly. Hence, I am happy to recommend the work for publication in Nature Communications.

Reviewer #4

Authors have fully addressed all the comments from the first round of review. I congratulate them for performing the excellent research presented in this manuscript.

Reviewer #4 (Remarks on code availability):

I tested the python notebooks and they worked correctly. The data and code for generating figures also looked fine. Authors could provide comments at the top of the .py files to explain the goal and the inputs/outputs of the code.

Reviewer #5

Reviewer #2

The authors have addressed some of the points raised. However, several issues require further attention. First notes on their replies to reviewers' comments.

R2-1: In their reply the authors mention 101 structures from independent 500 fs MLP MD trajectories. 500 fs is clearly too short to sufficiently sample conformational degrees of freedom. Either these are 500 ps or this analysis needs to be redone.

In the first round of revisions (response to Reviewer #1, comment no. 3), we addressed concerns regarding the duration of our MD trajectories for validating our MLPs. Specifically, we conducted a point-to-point validation over 3 ps of MD simulations using the trained MLP (Figure R3 in the revision, reported as Figure S13 (water) and S18 (methanol) in the SI), which fully covers the time frame of the reaction and solvent reorganization. Our analysis demonstrates that the performance of the model, measured by the MAD in energy and forces, is consistent with the validation performed over the 500 fs time frame. As our study focuses on a reactive event occurring

in the femtosecond timescale from the transition state (TS), carrying a validation over the 500 fs trajectory sufficiently covers the degrees of freedom for the system under study (see SI § S5 and S6). This conclusion is further supported by our downhill dynamics study and the depiction of the distribution of the training dataset along the reaction pathways (Figure R1 in the previous response), which confirms that the trained MLP adequately captures the conformational degrees of freedom from TS to reactant and product states. While longer simulation times will enhance the sampling at the reactant or product states, such extended sampling is not expected to improve the characterization of the systems as the molecules under study have low conformational flexibility and the solvent employed has a fast relaxation time. Therefore, we maintain that our methodology appropriately captures the dynamics and conformational changes relevant to the reactive event under investigation and that more extended testing may be needed for increasingly complex systems currently studied within our group; this is not the case here.

R2-2: In their reply letter they write “In the table below” – does this refer to Table R2 in the letter? (S7) probably refers to a newly added table in the SI? It would be useful to state this explicitly in the reply letter. The performance of the trained PES on true “test structures” (Table S7 entries 1 and 2) clarify that the quality of the trained NN is not particularly good. Therefore, the question still remains what the authors mean when writing “. . . the generated machine learning potentials exhibit excellent agreement with experimental data. . .” in the abstract and whether this agreement is not fortuitous given that their MLP does not generalize that well to out-of-sample structures.

The reference to “the table below” in our letter indeed refers to Table R2, where we present the errors for two sets of systems for *endo/exo* reactions in two different solvents. This table has now been included in the SI as Table S7, which is referenced in the main text in the section Training Strategy – DA reaction of CP and MVK in explicit water.

We would like to emphasise that our approach does not involve training a NN model, as referred by the Reviewer. Instead, we utilized the linear Atomic Cluster Expansion (ACE) methodology (G. Csanyi *et al.*, *J. Chem. Theory Comput.*, **2021**, 17, 7696-7711), which combines linear regression with the ACE descriptor.

Regarding the performance evaluation of our methodology, entries 1 and 2 of Table S7 correspond to the small microsolvated systems and show the largest errors. However, we would like to stress that these entries involve systems containing transition states with randomly placed water molecules and include a different number of solvent molecules than in the configurations used for training, representing extreme situations of distorted structures. These tests were conducted specifically to assess the extrapolation capabilities of our model, which we find satisfactory. For example, our model reaches energy errors for $\text{TS}_{\text{CP+MKV}} + 3 \text{H}_2\text{O}$ of *endo* and *exo* (Table S7 entries 1 and 2) of 2.07 and 1.77 meV/atom. Even considering the error on the entire system, the magnitude of the error is 40.63 and 37.93 meV, achieving the chemical accuracy of 43 meV (=1 kcal mol⁻¹). By conducting these tests, we gained insight into the limits of MLPs and identified situations involving

distorted and closely positioned solvent molecules where caution is required.

It is important to note that our model demonstrates significantly lower mean absolute deviations (MAD) in energy and forces for systems relevant to the presented study—reactions in fully solvated environments. These results provide a reliable description of chemical reactions in realistic conditions, which is the primary focus of our investigation.

In conclusion, we have rigorously tested our methodology, including its performance on out-of-sample structures, and we stand by the robustness of our approach. We appreciate that reporting those challenging cases is important and that is the reason why we have included them in this work.

R2-3: It appears that the MLP presented here is not really superior to DeepMD NN. Why would one then expect better agreement for experimental observables?

We believe there may be a misunderstanding regarding the aims of our study and the implications drawn from our comparisons. Firstly, we would like to clarify that the purpose of the study was not to develop a model with performance superior to DeepMD. Rather, we intended to demonstrate that accuracy comparable to state-of-the-art approaches can be obtained at a significantly lower computational cost. The work of Parinello et al. (*Catal. Today* 387 (2022) 143–149), exploring urea decomposition in water with a DeepMD NN, is used as an example to demonstrate that our model achieves comparable accuracy with significantly fewer data (i.e., 600 vs 14,536 data points for DeepMD). This underscores the relevance of our fully automated strategy and its potential to be used by the broader community. Furthermore, our approach offers distinct advantages compared to available training approaches, particularly in terms of computational efficiency. For example, by training our model only on cluster data as opposed to fully periodic data, we can employ high-level theory methods for training, which will be impractical using periodic ab initio MD to generate training sets. In this regard, our approach allows us to apply a range-separated hybrid DFT functional (ω B97M-D3BJ/def2-TZVP), which is significantly more accurate for reactivity modelling than traditional GGA DFT functionals, such as PBE/m-DZVP applied in the urea decomposition work. We firmly believe that combining the high level of theory, together with the model’s accuracy comparable to well-performing state-of-the-art NNs, provides strong grounds to expect that the agreement with experimental data comes from the improvement of the physical description of the reactive system as opposed to fortunate error cancellation.

R2-4: Given that the quality of the present MLP is not superior to a DeepMD NN hyperparameter optimization would be meaningful, not?

We understand the importance of hyperparameter optimization, especially in the context of NN models like DeepMD. In such models, hyperparameters, including the number of layers, nodes per layer, and learning rate, are crucial for model performance and demand careful adjustment. However, the model we employed, linear regression with regularization, draws on domain-specific knowledge to determine hyperparameters, making it less likely that hyperparameter optimization would markedly improve its performance. Indeed, the selection of most hyperparameters in the

linear ACE is based on chemical and physical knowledge, as well as previous testing in related studies (see, for example, *Phys. Rev. B*, 99, **2019**, 014104 and *J. Comput. Phys.*, 454, **2022**, 110946 for further discussion). For instance, the hyperparameter ν , which represents the maximum correlation number, is informed by a deep understanding of the underlying physics of the model rather than arbitrary tuning. The chosen value of $\nu = 4$ ensures the inclusion of all relevant body interactions up to five bodies, which, based on previous validation (*Phys. Rev. B*, 99, **2019**, 014104; *Phys. Rev. Materials*, 6, **2022**, 013804), is optimal for the system under study.

R2-5: Downhill trajectories assume that the reaction passes through the transition state. This is not necessarily true, e.g. recent work by Käser et al on malonaldehyde. The problem accentuates in going from the gas phase to solution because depending on the solvent structure the TS may change. In addition, downhill trajectories assume that the system crosses the TS with near-zero velocity which is also not necessarily the case. It is not good enough to state that such procedures have been “widely employed”. The authors should run a forward simulation in gas phase and in solution to see how the system behaves.

We appreciate the reviewer’s comments about this classical method and their concerns about its performance. In the manuscript, Figure S26 and S27 in SI § S10.1, we plotted the trajectories corresponding to all downhill trajectories considered for analysis of the reaction in the explicit solvent. The figures show that the simulations start at the TS point and finish in the reactant or product state, in agreement with the static PES scan. Furthermore, the 2D scans identified the zwitterionic regions where only one bond is formed, indicating the change of stability of these species in different environments and aligning with the change of the synchronicity of the reaction. We would also like to point out that the Gibbs free energy barriers were obtained from Umbrella sampling (US) simulations, where the applied bias ensured proper sampling in the given region of the free energy surface, including TS. US simulations and downhill dynamics also both confirm the existence of the entropic intermediates in explicit water. The results from our downhill dynamics are, therefore, fully consistent with other types of analysis performed.

R2-6: The authors should validate their MLP on the systems sizes for which DFT calculations are possible. If I understand correctly, this is MVK+CP in 55 water and 40 methanol molecules (as per the reply to R2-6).

We find this comment unclear, as we have already validated the potential of these systems. As detailed in our original manuscript, specifically in Table R2, our validation, using point-to-point energy and forces, indeed encompasses a range of system sizes, including the configurations involving MVK+CP in environments of 55 water and 40 methanol molecules. These systems were chosen to illustrate the predictive capability of our MLPs in systems not included in training and where DFT computations remain feasible.

A few additional questions:

1. The authors should demonstrate that their trajectories conserve energy.

Section S3.1 Water models in the original version of the SI contains the analysis of NVE dynamics for the bulk water system, confirming the energy conservation in the dynamics using ACE potential. Here, we extend this analysis also for the reactive system, i.e., *endo* reaction in 320 water molecules shown in Fig. S1. The NVE dynamics is initiated from the reactant state in equilibrated water. This analysis of the trajectory demonstrates that the energy is conserved throughout the 50 ps NVE dynamics.

Figure S1: Energy fluctuations for ACE MLP of the *endo* reaction in 320 water throughout the 50 ps NVE simulation under PBC.

2. It is unclear where the experimental activation free energies of 19.2 and 21.1 kcal/mol come from. Ref. 57 deals with the reaction in nonaqueous polar solvents.

The Ref. 57 indeed focuses on "Diels Alder reactions in nonaqueous polar solvents"; however, the first table in the publication lists experimental values of the second-order rate constant of reaction of CP and MVK in various solvents, including water. The source of these data is referenced as *J. Am. Chem. Soc.*, 102, 26, **1980**, 7816–7817. Kinetic experimental data in water are also reported in *Pure & App. Chem.*, 67, 5, **1995**, pp. 823-828. While these values are not directly reported in terms of the free energy, several other publications, such as *J. Am. Chem. Soc.* 7, 115, **1993**, 2936-2942; *J. Chem. Theory Comput.* 3, **2007**, 1412–1419; *J. Am. Chem. Soc.* 132, **2010**, 3097-3104; *J. Chem. Theory Comput.* 12, **2016**, 4735-4727; *J. Phys. Chem. B* 123, **2019**, 5131-5138, reports these values in terms of experimental ΔG^\ddagger and $\Delta\Delta G^\ddagger$ energies. To clarify this, we added *Pure & App. Chem.*, 67, 5, **1995**, pp. 823-828, and *J. Am. Chem. Soc.* 132, (**2010**), 3097-3104 as reference 58 and 66.

3. What value for the dielectric constant was used for the simulations in implicit water?

The implicit solvent computations were performed using the CPCM model in ORCA 4.2.1 using default parameters, including dielectric constant values of 80.4 for water and 32.63 for methanol.

4. Is DFT really good enough to distinguish zwitterionic from biradical character of the reaction in solvent?

While numerous publications have employed DFT to investigate the zwitterionic or biradical nature of reactions in the gas phase (*Angew. Chem.*, 131, **2019**, 6481-6485; *J. Comput. Chem.*, 40, **2019**, 854–865; *J. Am. Chem. Soc.*, 137, **2015**, 3975–3980; *Chem. Sci.*, 7, **2016**, 745-751 with experimental data as validations), there is a lack of literature on distinguishing between the zwitterionic and biradical character of species in solution. The thorough benchmark of the DFT against SCS-MP2 reference strongly supports the use of selected DFT methods to model the DA reaction. Furthermore, in none of our calculations have we seen any spin contamination or another descriptor that indicates the use of a single reference method is insufficient. Since the quality of the MLP is inevitably linked to the performance of the DFT method, we conclude that the species observed in the dynamics have a zwitterionic character, independent of DFT’s ability to distinguish between these characteristics. To clarify this in the manuscript, we modified the following part in the SI § S8):

To characterise the electronic structure of the species, we calculated the total spin expectation value $\langle S^2 \rangle$ of structures in this region with U ω B97M-D3BJ/def2-TZVP, with multiplicity set to 1. The value of $\langle S^2 \rangle$ converged to zero, demonstrating the absence of diradical character and confirming the formation of zwitterionic species (ZS). Furthermore, we did not observe any spin contamination which would indicate a complicated electronic structure. As the ACE MLP were trained solely on energy and forces, they cannot provide direct information on the electronic structure. Instead, the species formed in the MLP dynamics reflect the stability at the DFT level.

REVIEWER COMMENTS

Reviewer #2 (Remarks to the Author):

Re-review of Duarte et al.

The authors have provided reasonings and replies to the points raised which, however, do not address the fundamental issues.

1. The accuracy of the trained model is not sufficiently high to warrant "excellent agreement" as stated in the abstract. As an example, see Figure S13 right upper panel which reports a MAD of 0.17 eV. Neither DFT-based reference energies nor model performance of 0.17 eV are sufficient to expect "excellent agreement" with experiment. If there is "agreement" it is rather fortuitous.

2. The simulations are overall too short and simulations starting at the TS are misleading. In general, reactive trajectories originating from the reactant side and developing in an unconstrained fashion will rarely sample the (gas phase) TS. While trajectories starting from the TS have been used earlier in the literature (which does not make them more meaningful, though..) and may provide a basis for some mechanistic reasoning, they should not be "sold" as how the reaction actually proceeds. It is also noted that the time scale of the simulations is clearly insufficient to sample solvent reorganization on the diffusion time scale with is 10s of picoseconds or longer. Hence, stating that the simulations "fully cover...solvent reorganization" (reply letter) is simply wrong.

At this stage the claim of "excellent agreement" is not supported by the data presented and should be removed.

Reviewer #2

1. The accuracy of the trained model is not sufficiently high to warrant "excellent agreement" as stated in the abstract. As an example, see Figure S13 right upper panel which reports a MAD of 0.17 eV. Neither DFT-based reference energies nor model performance of 0.17 eV are sufficient to expect "excellent agreement" with experiment. If there is "agreement" it is rather fortuitous.

In our original submission and subsequent responses, we have provided a solid benchmark against SCS-MP2, a reference method recommended in previous benchmark studies for similar systems (Angew. Chem., Int. Ed., 47, 2008, 7746-7749 and Chem.–Eur. J., 10, 2004, 6468-6475). Our results and those from independent benchmarks demonstrate the good performance of the ω B97M-D3BJ functional against this level of theory as evidenced by section S4 in Supporting Information (table S5 and figure S6). Indeed, this function ranks among the accurate and reliable range-separated hybrid meta-GGA functionals (J. Chem. Theory Comput., 14, 2018, 5725–5738). Furthermore, when combined with the triple zeta basis set, our methodology surpasses the typical level of theory for modelling reactions in explicitly solvated environments using ab initio MD simulations.

Regarding the concerns raised about the accuracy of our models, The errors reported in our work need to be evaluated in relation to the overall energy scale of the studied systems, which is approximately -128124.04 eV. This means that the absolute error corresponds to approximately 0.0001 %, translating to 0.91 meV/atom. This error is comparable to reported ML models for reactions in complex solvated systems, see Catal. Today, 387, 2022, 143–149, where the reported error on the testing set is of the same order, i.e. 6 kJ/mol (0.57 meV/atom). While we are confident about the quality of our results, we respect the reviewer's scepticism and alter the formulation in the abstract and discussion section.

2. The simulations are overall too short and simulations starting at the TS are misleading. In general, reactive trajectories originating from the reactant side and developing in an unconstrained fashion will rarely sample the (gas phase) TS. While trajectories starting from the TS have been used earlier in the literature (which does not make them more meaningful, though..) and may provide a basis for some mechanistic reasoning, they should not be "sold" as how the reaction actually proceeds. It is also noted that the time scale of the simulations is clearly insufficient to sample solvent reorganization on the diffusion time scale which is 10s of picoseconds or longer. Hence, stating that the simulations "fully cover...solvent reorganization" (reply letter) is simply wrong. At this stage the claim of "excellent agreement" is not supported by the data presented and should be removed.

Our downhill dynamics approach does not exclusively initiate from the transition state (TS) structure. Rather, we sample within the TS region to consider the impact of solvent. In addition, we use other uphill dynamics and umbrella sampling, which are also consistent with our downhill dynamics. All these approaches indicate that the reaction proceeds within the sampling timescale, making longer simulations unnecessary. Sampling "solvent reorganization on the diffusion time scale" is not relevant here as the reorganisation of the water molecules happens within less than five ps (Proc. Natl. Acad. Sci. U.S.A., 104, 43, 2007, 16731-16738, J. Phys. Chem. B, 115, 18, 2011, 5604–5616) and diffusion of solvent from the reactant/product does not influence the chemical process, although we have already shown that our potentials are stable for hundred picoseconds. Overall, our aim has been to demonstrate that our data efficiency training strategy can yield stable machine learning potential to model reactions in explicit solvents with comparable accuracy to the reference method. Consistency in our results has been shown through various dynamic analyses including downhill dynamics, uphill dynamics, and umbrella sampling.

REVIEWERS' COMMENTS

Reviewer #2 (Remarks to the Author):

The authors have addressed and replied satisfactorily to the remaining points raised and the work is recommended for publication.